# Identification of the factor XII contact activation site enables sensitive coagulation diagnostics

Marco Heestermans [1,14], Clément Naudin [1,2,3,14], Reiner K. Mailer[1,14], Sandra Konrath[1], Kristin Klaetschke[1], Anne Jämsä[4], Maike Frye [1], Carsten Deppermann[1], Giordano Pula[1], Piotr Kuta[1], Manuel A. Friese [5], Mathias Gelderblom[6], Albert Sickmann[7], Roger J. S. Preston[8], Jerzy-Roch Nofer[1], Stefan Rose-John [9], Lynn M. Butler[1,2,3], Ophira Salomon[10], Evi X. Stavrou [11,12] & Thomas Renné [1,13 ✉]

Contact activation refers to the process of surface-induced activation of factor XII (FXII), which initiates blood coagulation and is captured by the activated partial thromboplastin time (aPTT) assay. Here, we show the mechanism and diagnostic implications of FXII contact activation. Screening of recombinant FXII mutants identified a continuous stretch of residues Gln317–Ser339 that was essential for FXII surface binding and activation, thrombin generation and coagulation. Peptides spanning these 23 residues competed with surface-induced FXII activation. Although FXII mutants lacking residues Gln317–Ser339 were susceptible to activation by plasmin and plasma kallikrein, they were ineffective in supporting arterial and venous thrombus formation in mice. Antibodies raised against the Gln317–Ser339 region induced FXII activation and triggered controllable contact activation in solution leading to thrombin generation by the intrinsic pathway of coagulation. The antibody-activated aPTT allows for standardization of particulate aPTT reagents and for sensitive monitoring of coagulation factors VIII, IX, XI.

[1] Institute of Clinical Chemistry and Laboratory Medicine, University Medical Center Hamburg-Eppendorf, Hamburg, Germany. [2] Department of Clinical Medicine, The Arctic University of Norway, Tromsø, Norway. [3] Clinical Chemistry and Coagulation Research, Department of Molecular Medicine and Surgery, Karolinska Institute, Stockholm, Sweden. [4] Karolinska University Hospital, Stockholm, Sweden. [5] Institute for Neuroimmunology and Multiple Sclerosis, University Medical Center Hamburg-Eppendorf, Hamburg, Germany. [6] Department of Neurology, University Medical Center Hamburg-Eppendorf, Hamburg, Germany. [7] Leibniz-Institut für Analytische Wissenschaften - ISAS - e.V., Dortmund, Germany. [8] Irish Centre for Vascular Biology, School of Pharmacy and Biomolecular Sciences, Royal College of Surgeons in Ireland, Dublin, Ireland. [9] Institute of Biochemistry, University of Kiel, Kiel, Germany. [10] Institute of Thrombosis and Hemostasis, Sheba Medical Center, Tel Hashomer, Israel and Sackler Faculty of Medicine, University of Tel Aviv, Tel Aviv, Israel. [11] Department of Medicine, Louis Stokes Veterans Administration Medical Center, Cleveland, OH, USA. [12] Department of Medicine, Hematology and Oncology Division, Case Western Reserve University School of Medicine, Cleveland, OH, USA. [13] Center for Thrombosis and Hemostasis (CTH), Johannes Gutenberg University Medical Center, Mainz, Germany. [14]These authors contributed equally: Marco Heestermans, Clément Naudin, Reiner K. Mailer. ✉email: thomas@renne.net

**B**lood coagulation is essential to terminate bleeding at sites of vascular injury (haemostasis). However, uncontrolled coagulation contributes to vascular occlusion (thrombosis) that remains the primary cause of morbidity and mortality in the Western world[1].

Blood coagulation can be initiated by two principal mechanisms, namely the extrinsic and intrinsic coagulation pathways. The extrinsic pathway is triggered by the transmembrane protein tissue factor (TF) exposed on subendothelial tissues at sites of vascular injury. In contrast, the intrinsic pathway of coagulation commences intravascularly through activation of circulating coagulation factor XII (FXII, Hageman Factor, EC:3.4.21.38), in a reaction that involves high molecular weight kininogen (HK) and plasma prekallikrein (PK). These three factors are collectively referred to as the plasma contact activation system, a name that originates from the unique mode of FXII activation. FXII circulates in plasma in a zymogen form, however, binding ("contact") to negatively charged surfaces induces a conformational change in zymogen FXII leading to proteolytic activity and formation of the activated protease FXIIa ("contact activation"). FXIIa activation of PK forms plasma kallikrein (PKa) that reciprocally activates FXII, thereby amplifying FXIIa formation in solution ("fluid phase activation")[2]. FXIIa contributes to coagulation via its substrate coagulation factor XI (FXI)[3–5] and targeting the FXIIa protease in experimental murine[6] and large animal[7] models provides protection from thrombosis (reviewed in[4,8–10]) indicating a critical role of FXIIa-triggered coagulation in thrombus formation. Since FXIIa impacts thrombosis but does not influence haemostasis, its pharmacologic inhibition has emerged as a potential target for thrombosis prevention without bleeding risk[11,12].

The activated partial thromboplastin time (aPTT) coagulation test is diagnostically used to monitor the integrity of the intrinsic pathway of coagulation[13] and is performed >5 billion times annually. The aPTT assay relies on FXII contact activation triggered by negatively charged non-physiological particles, such as the white clay material kaolin, micronized silica or the organic compound ellagic acid. Surface properties of FXII activating particles have remained poorly characterized, resulting in challenges when standardizing aPTT-based diagnostics across various contact-activating reagents[14,15]. Recently, several substances such as collagen, RNA, neutrophil extracellular traps (NETs)[16], polyphosphate (polyP, an inorganic polymer that forms $Ca^{2+}$-rich nanoparticles that are retained on activated platelet surfaces[17,18]) and medical materials such as intravascular catheters[19] have been identified as FXII contact activators in vivo with implications for thrombus formation[20,21]. Despite this growing list of potentially relevant FXII contact activators and their role in thrombosis, it remains unknown how FXII recognizes surfaces leading to zymogen contact activation.

In the current study, we screened FXII for sequences required for contact activation and found that a continuous stretch of 23 amino acids (Gln317-Ser339, designated PR-III) within the FXII proline-rich domain is essential for surface-induced FXII activation and coagulation. Similarly, a recombinant PR-III derived peptide competed with full-length FXII for surface-triggered activation. In contrast to full-length FXII, mutants lacking the PR-III region were defective in sustaining thrombosis in mice. Antibodies raised against PR-III recapitulated the effects of a surface and induced FXII activation in a regulated manner and fully in solution. Based on these findings, we used the FXII activating antibodies to establish sensitive coagulation assays that offer novel diagnostic methods beyond the range of application of current particle-driven aPTT tests. Together, the present study provides insight into the mechanism of FXII contact activation and translates the structural findings into the development of novel diagnostics with implications for improved patient care.

## Results

**Generation of FXII heavy chain mutants.** FXII is a multidomain protein and its heavy chain is composed of a fibronectin type-II (Fib-II) domain, the first epidermal growth factor-like (EGF-I) domain, a fibronectin type-I (Fib-I) domain, a second EGF-like (EGF-II) domain, a kringle (KR) domain and a prolinerich (PR) domain, while the serine protease domain is located in the light chain. The FXII heavy chain mediates contact to surfaces[22]. To screen for sequences within the human FXII heavy chain involved in contact activation, we generated recombinant full-length FXII (FXII_fl) and 19 mutants lacking a single or a combination of domains and domain fragments of the heavy chain (Fig. 1A). HEK293 cells expressing FXII_fl and FXII deletion variants migrated at the calculated apparent molecular weights in western blot analyses of lysed transfected cells and their supernatants (Fig. 1B, C). The N-terminal endogenous FXII signal peptide mediated secretion of 15/20 constructs. However, FXII_ΔEGF-II, FXII_ΔPR, FXII_ΔEGF-II–PR, FXII_ΔEGF-II-1 and FXII_ΔEGF-II-2 variants were retained intracellularly and excluded from further studies. To confirm these results in alternative cell lines, we used Chinese hamster ovary (CHO) cells and HepG2 hepatocytes. Both CHO and HepG2 cells expressed FXII mutants which migrated at similar apparent molecular weights as HEK293-derived proteins. Moreover, FXII mutants that were defective in secretion from HEK293 cells, were similarly not secreted from CHO and HepG2 cells (Supplementary Fig. 1A, B).

**FXII mutants lacking the PR-III region exhibit defective contact activation.** To identify sequences critical for contact activation, we analysed FXII mutants for their ability to restore defective surface-induced coagulation of FXII-deficient human plasma (Fig. 2A). As expected, the kaolin-driven aPTT clotting time was largely prolonged in FXII-deficient plasma (no detectable FXII antigen by western blotting) supplemented with buffer control ($427 \pm 9$ s). Reconstituting FXII-deficient plasma with plasma-derived FXII (FXII_pd) or recombinant FXII_fl to physiologic FXII levels (30 μg/mL, 375 nM), normalized the aPTT to similar levels recorded for normal platelet poor plasma (PPP; $51 \pm 18$ and $49 \pm 7$ vs. $44 \pm 6$ s, respectively). Similar to FXII_fl, reconstituting FXII-deficient plasma with FXII mutants FXII_ΔFib-II, FXII_ΔEGF-I, FXII_ΔFib-I, FXII_ΔKR or FXII_ΔFib-II/EGF-I restored the prolonged clotting time, indicating that Fib-I, EGF-I, Fib-II and KR domains are dispensable for zymogen contact activation. In contrast, reconstitution of FXII-deficient plasma with FXII deletion mutants lacking the proline-rich (PR) domain, failed to rescue the defective surface-triggered clotting (FXII_ΔHC, $303 \pm 2$ s; FXII_ΔEGF-I − PR, $182 \pm 16$ s or FXII_ΔKR/PR, $211 \pm 22$ s). To map the site mediating FXII contact activation, we tested five FXII mutants lacking the N-terminal (Pro279-His293, FXII_ΔPR-I), central (Val294-Ser316, FXII_ΔPR-II) or C-terminal (Gln317-Ser339, FXII_ΔPR-III) portion of the PR domain and combinations thereof (FXII_ΔPR-I/PR-II and FXII_ΔPR-II/PR-III). Addition of FXII_ΔPR-I, FXII_ΔPR-II or FXII_ΔPR-I/PR-II to FXII-deficient plasma led to normalization of aPTT values ($50 \pm 3$, $65 \pm 14$ and $48 \pm 7$ s, respectively). In contrast, variants lacking PR-III (FXII_ΔPR-II/PR-III and FXII_ΔPR-III) were inactive and did not correct the aPTT time of FXII-deficient plasma ($198 \pm 24$ and $442 \pm 79$ s, respectively).

We used fluorometric real-time thrombin generation assays as an independent method to test for FXII sequences required for contact activation (Fig. 2B–E). Consistent with previous data[23,24], kaolin-stimulated thrombin production was undetectable in FXII-deficient plasma spiked with buffer (endogenous thrombin potential, ETP $100 \pm 10$ nM*min). Impaired thrombin formation

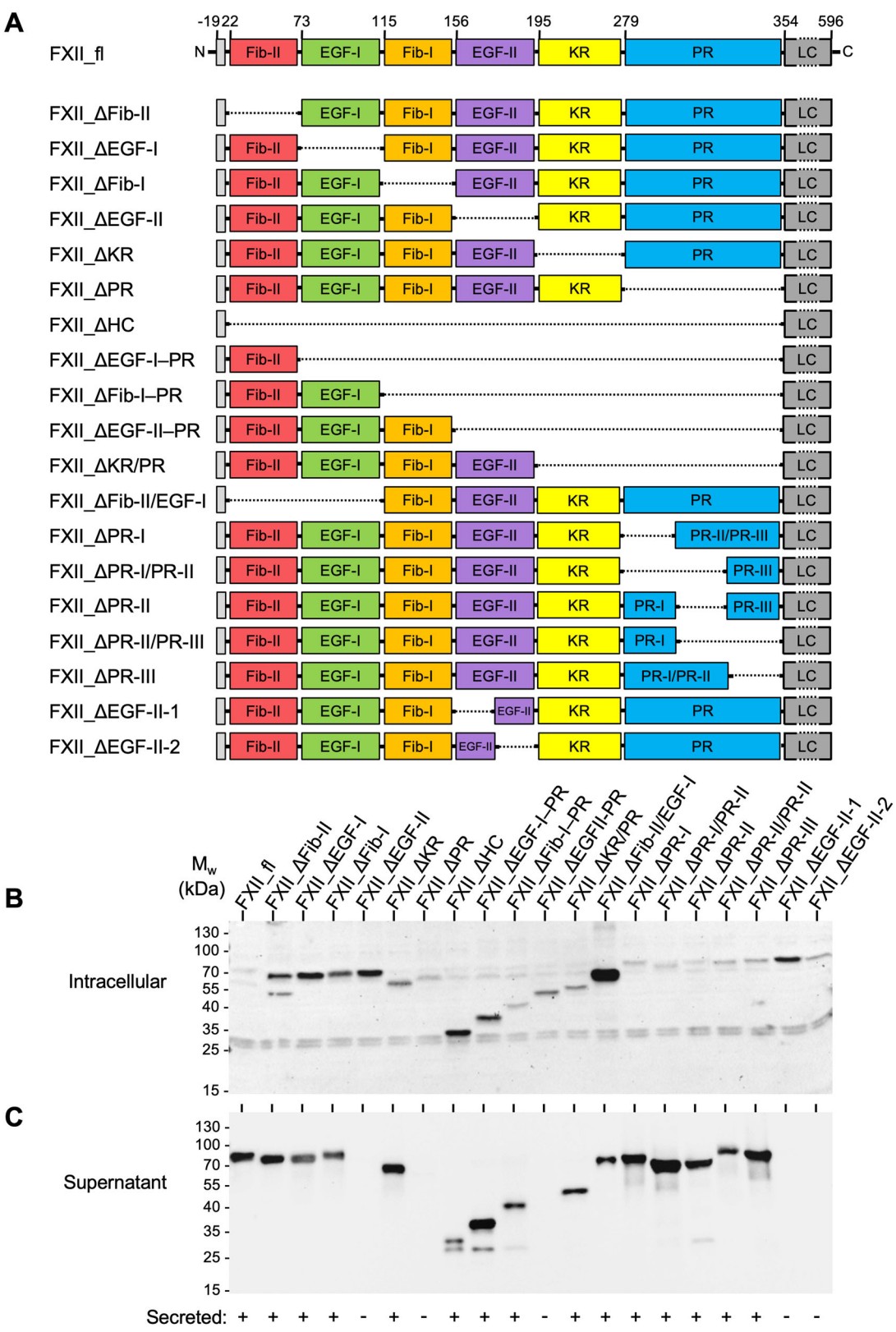

was restored following the addition of FXII_pd or FXII_fl (1441 ± 8 and 1555 ± 32 nM*min, respectively) to levels comparable to thrombin generation in normal PPP (1548 ± 51 nM*min). However, and in line with kaolin-triggered clotting results, reconstituting FXII-deficient samples with FXII mutants lacking the PR-III site, alone or in combination with other domain fragments (FXII_ΔPR-III, FXII_ΔHC, FXII_ΔEGF-I–PR,

FXII_ΔKR/PR or FXII_ΔPR-II/PR-III), failed to restore thrombin generation (<350 nM*min). In contrast, addition of mutants lacking the Fib-I, Fib-II, EGF-I or KR domains (FXII_ΔFib-II, FXII_ΔEGF-I, FXII_ΔFib-I, FXII_ΔKR or FXII_ΔFib-II/EGF-I) or N- and central portions of the PR domain (FXII_ΔPR-I, FXII_ΔPR-II or FXII_ΔPR-I/PR-II) were able to restore thrombin formation in FXII-deficient plasma (>1400 nM*min; Fig. 2B, D).

**Fig. 1 FXII deletion mutants. A** Schematic structure of the zymogen forms of human full-length FXII (FXII_fl) and FXII deletion mutants lacking various domains or fragments. The N-terminal grey square represents the 19 amino acid-long signal peptide. The heavy chain of mature FXII consists of six domains: the fibronectin type-II (Fib-II, red), the first epidermal growth factor like (EGF-I, green), the fibronectin type-I (Fib-I, orange), the second EGF-like (EGF-II, violet), the Kringle (KR, yellow), and the proline-rich (PR, blue) domains. The C-terminal light chain (LC) contains the catalytic triad of a serine protease and is marked in grey. PR-I, PR-II and PR-III represent the N-terminal, central and C-terminal fragment of the PR domain, respectively. FXII_ΔEGF-II-1 and FXII_ΔEGFII-2 are mutants deficient in the N- and C-terminal portions of the EGF-I domain, respectively. Numbers on top indicate amino acid residues of the mature FXII protein. **B, C** HEK293 cells were transiently transfected with vectors coding for FXII mutants shown in panel A. After 48 h of transfection, washed HEK293 cells (**B**) or cell supernatants (**C**) were analysed for FXII protein using SDS PAGE under reducing conditions and western blotting with polyclonal anti-FXII antibodies. Representative images of n = 5 individual experiments. In the bottom panel, + or – indicate whether mutants were secreted into the cell supernatant. $M_w$: molecular weight, kDa: kilodalton.

Consistently, reconstituting FXII-deficient plasma with CHO-expressed FXII_fl, but not with FXII_ΔPR-III, restored defective thrombin formation in response to kaolin and ellagic acid (Supplementary Fig. 1C, D).

We next determined surface-triggered thrombin formation by a physiologic contact activator. For these studies, we utilized long-chain polyP, an endogenous FXII activator[17,25,26]. We found that in FXII-deficient plasma supplemented with FXII_fl, FXII_ΔPR-I, FXII_ΔPR-II or FXII_ΔPR-I/PR-II variants, polyP led to robust thrombin production that was comparable to levels generated in normal PPP (>750 nM*min) (Fig. 2C). In contrast, polyP-induced thrombin generation in FXII-deficient plasma supplemented with FXII_ΔPR-III or FXII_ΔPR-II/PR-III variants, was as low as in buffer-spiked samples (143 ± 25 and 128 ± 12 vs. 100 ± 10 nM*min, P > 0.05; Fig. 2C, E). Reconstitution of FXII-deficient plasma with FXII_fl led to robust thrombin generation induced by E. coli-derived polyP and ellagic acid. In contrast, coagulation triggered by polyP and ellagic acid in buffer or FXII_ΔPR-III spiked FXII-deficient plasma was defective (Supplementary Fig. 2A–D). These combined studies indicate that the PR-III region is required for FXII contact activation and FXII-induced coagulation in plasma.

**FXII_ΔPR-III is defective in contact activation but susceptible to fluid phase activation**. In addition to surface-mediated activation of zymogen FXII ("contact activation"), plasma proteases such as PKa and plasmin can activate FXII in solution ("fluid phase activation"). In this framework, we next evaluated if the PR-III region is involved in protease-mediated FXII activation. FXII_fl or FXII_ΔPR-III mutants were incubated with contact activators kaolin (Fig. 3A, B), short-chain polyP (Fig. 3C, D), dextran sulphate [DXS, (Fig. 3E, F)] or PKa (Fig. 3G, H). Time-dependently, kaolin, polyP and DXS induced FXII_fl cleavage, as seen by the disappearance of the 80 kDa zymogen band and the concomitant appearance of the heavy chain FXIIa fragment (50 kDa) by immunoblotting. In contrast, even after 240 min incubation with kaolin, polyP or DXS, FXII_ΔPR-III remained intact and no heavy or light chain fragments were detected. Distinctly, PKa readily cleaved FXII_ΔPR-III even faster than its cleavage of FXII_fl. Following a 15 min incubation, PKa activated the FXII_ΔPR-III, as indicated by the appearance of a heavy chain fragment. In contrast, PKa did not significantly activate FXII_fl under these conditions. FXIIa generation was also measured by chromogenic substrate S2302 hydrolysis (Fig. 3I, J; Supplementary Fig. 2E, F). FXII_fl or FXII_ΔPR-III variants incubated with buffer control did not exhibit significant amidolytic activity. Consistent with immunoblotting and coagulation assays, chromogenic signal increased over time when FXII_fl, but not the FXII_ΔPR-III variant, were incubated with kaolin or DXS. In contrast, PKa (Fig. 3J), plasmin, and thrombin (Supplementary Fig. 2G, H) activated both FXII_fl and FXII_ΔPR-III, and consistent with published data, thrombin-mediated FXII_fl activation was low, as compared to PKa and plasmin[27]. Collectively, these

data indicate that the PR-III region is indispensable for surface-induced activation of FXII but deletion of PR-III still allows for protease-mediated FXII activation.

**The PR-III region mediates FXII binding to surfaces**. We next asked if the PR-III region alone can interfere with FXII contact activation by competing with full-length zymogen FXII for surface binding. We cloned and expressed the N-terminal (MBP-PR-I, Pro279-His293), central (MBP-PR-II, Val294-Ser316) and C-terminal (MBP-PR-III, Gln317-Ser339) parts of the PR domain as fusion proteins with maltose-binding protein (MBP, Fig. 4A). Coomassie brilliant blue staining (Fig. 4B, top panel) and immunoblotting using anti-MBP antibodies (Fig. 4B, bottom panel) confirmed the purity of recombinant proteins that migrated at the expected apparent molecular weight in SDS-PAGE. PR domain fragments were analysed for their interference with long-chain polyP-triggered coagulation in normal PPP. Consistent with previous data[25], polyP promoted thrombin formation in normal PPP. Pre-incubation of polyP with MBP, MBP-PR-I or MBP-PR-II did not interfere with surface-driven coagulation. In contrast, pre-incubation with MBP-PR-III reduced polyP-triggered thrombin formation by >2.5 fold as compared to MBP, MBP-PR-I or MBP-PR-II (ETP of 747 ± 5 vs. 2099 ± 56, 1749 ± 80 and 2050 ± 155 nM*min; Fig. 4C, D). In line with this inhibitory effect, MBP-PR-III also reduced polyP-triggered peak thrombin (41.0 ± 0 vs. 297.1 ± 3.8 nM) and prolonged the lag time and time to peak thrombin formation (15.6 ± 0.2 vs. 4.5 ± 0 min and 25.1 ± 0.2 vs. 7.9 ± 0 min, respectively; Supplementary Fig. 3A–C), compared to pre-incubation with MBP. In contrast, MBP-PR-III did not interfere with TF and FXa triggered thrombin formation (Supplementary Fig. 3D, E) or FXIIa-initiated plasma clotting (47 ± 6, 49 ± 7 and 45 ± 6 s in MBP-PR-III, MBP or buffer spiked normal PPP). Thrombin inhibitor dabigatran erased polyP-initiated coagulation, independently of the presence or absence of MBP-PR-III arguing against PR-III activities beyond coagulation (Supplementary Fig. 3F). To demonstrate a direct interaction between FXII and a surface, we performed a pulldown assay and found that FXII_fl, but not FXII_ΔPR-III, directly bound to kaolin (Fig. 4E). Furthermore, we confirmed a direct interaction of PR-III with immobilized long-chain polyP surfaces by an enzyme-linked immunosorbent assay (ELISA, Fig. 4F).

**FXII_ΔPR-III interferes with platelet-driven thrombosis in vivo**. We next examined the role of FXII PR-III in thrombosis in vivo. We induced thrombus formation in mesenteric arterioles by topical application of ferric chloride ($FeCl_3$) in mice. $FeCl_3$-induced thrombosis depends on platelet polyP and FXII gene-deleted ($F12^{-/-}$) mice are largely protected from occlusive thrombus formation in this model[17,28,29]. FXII_fl or FXII_ΔPR-III was intravenously (i.v.) injected in $F12^{-/-}$ mice to reconstitute FXII plasma levels to physiologic levels. Wild type (WT) mice served as control and in these mice, 6/6 $FeCl_3$-treated vessels occluded after

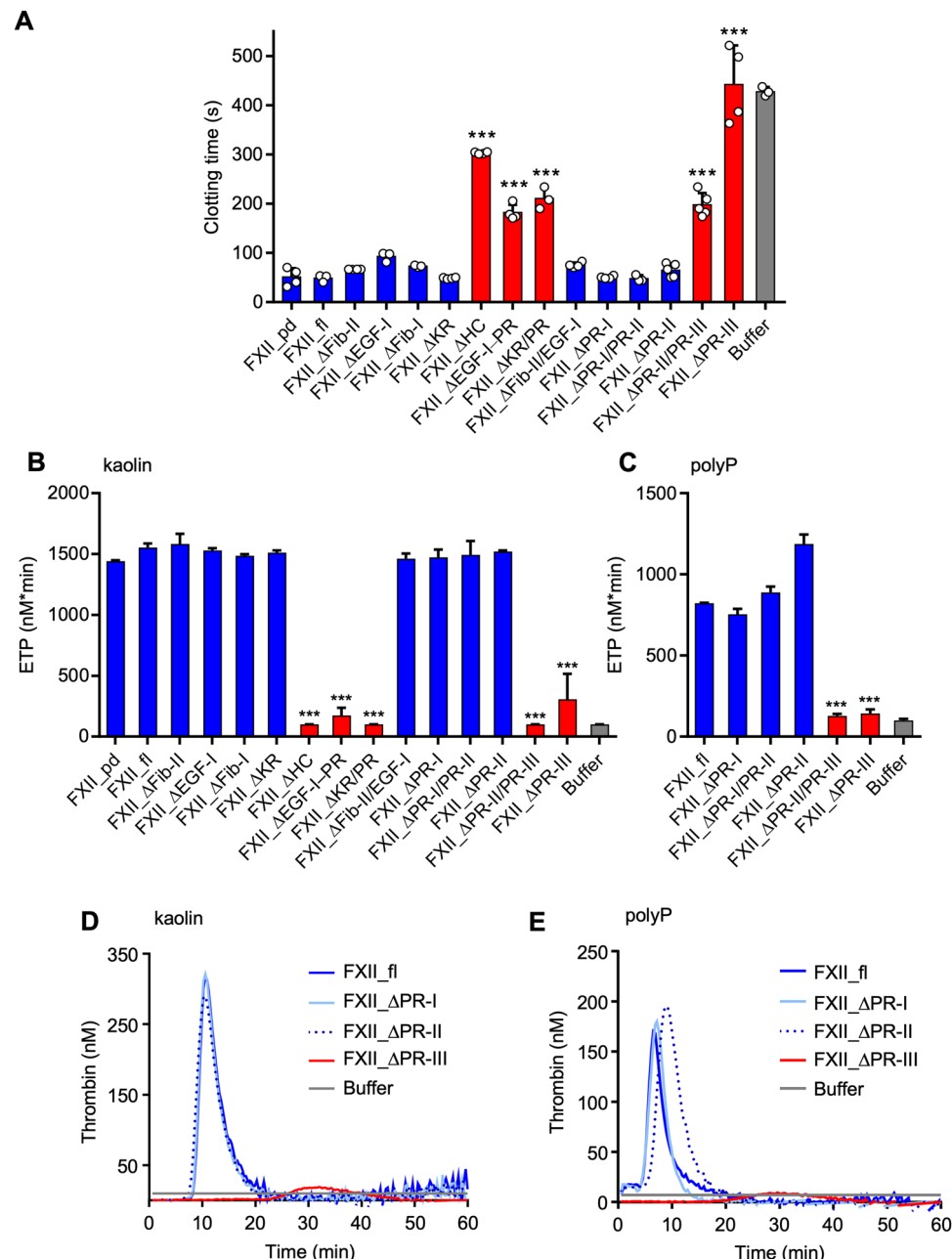

**Fig. 2 Contact activation of FXII mutants lacking PR-III is defective. A** Clotting times of kaolin-activated and recalcified FXII-deficient plasma, reconstituted with plasma-derived FXII (FXII_pd), recombinant full-length (FXII_fl), or recombinant FXII deletion mutants (375 nM each) or buffer, $n = 3$ independent clotting experiments. **B** Total thrombin formation (endogenous thrombin potential, ETP) of kaolin-activated FXII-deficient plasma, reconstituted with FXII_pd, recombinant FXII_fl, recombinant FXII deletion mutants (375 nM each) or buffer, assessed by real-time thrombin formation assays, $n = 3$. **C** ETP in long-chain polyP-activated (10 μg/mL) FXII-deficient plasma, reconstituted with FXII_pd, recombinant FXII_fl, FXII deletion mutants (375 nM each) or buffer. $n = 3$ real-time thrombin formation assays. Data in A–C are presented as mean values ± SD. ***: $P < 0.001$ vs. FXII_fl by one-way ANOVA and Dunnett´s multiple comparison test in **A–C**. **D, E** Representative real time thrombin generation curves of FXII-deficient plasma reconstituted with FXII_fl, FXII_ΔPR-I, FXII_ΔPR-II, FXII_ΔPR-III or buffer prior to activation with kaolin (**D**) or long-chain polyP (**E**).

$19 \pm 5$ min (Fig. 5A, B). Consistent with previous data[30,31], thrombus formation was defective in $F12^{-/-}$ mice, and all 6 vessels analyzed remained patent within the 40 min observation period. In FXII_fl-reconstituted $F12^{-/-}$ mice, platelets rapidly adhered to the injured vessel wall and occlusive thrombi developed in 6/6 vessels $18 \pm 4$ min post injury. In contrast, FXII_ΔPR-III-reconstituted $F12^{-/-}$ mice were significantly protected from vascular occlusion and occlusive thrombi formed in only 1 out of 6 vessels ($P < 0.05$ vs. FXII_fl reconstituted $F12^{-/-}$ mice).

To study surface-mediated contact activation in the venous system, we utilized a model of platelet-driven lethal pulmonary thromboembolism (PE). Platelets were activated by intravenous infusion of collagen/epinephrine and mice that survived > 30 min were considered survivors (Fig. 5C, lower panel). In line with earlier data[30,32], $F12^{-/-}$ mice were largely protected from precipitate disease in this PE model and 5/5 animals survived. However, all WT and $F12^{-/-}$ mice reconstituted with FXII_fl (5/5 each) universally succumbed with average survival times of $10.2 \pm 2.6$ min and

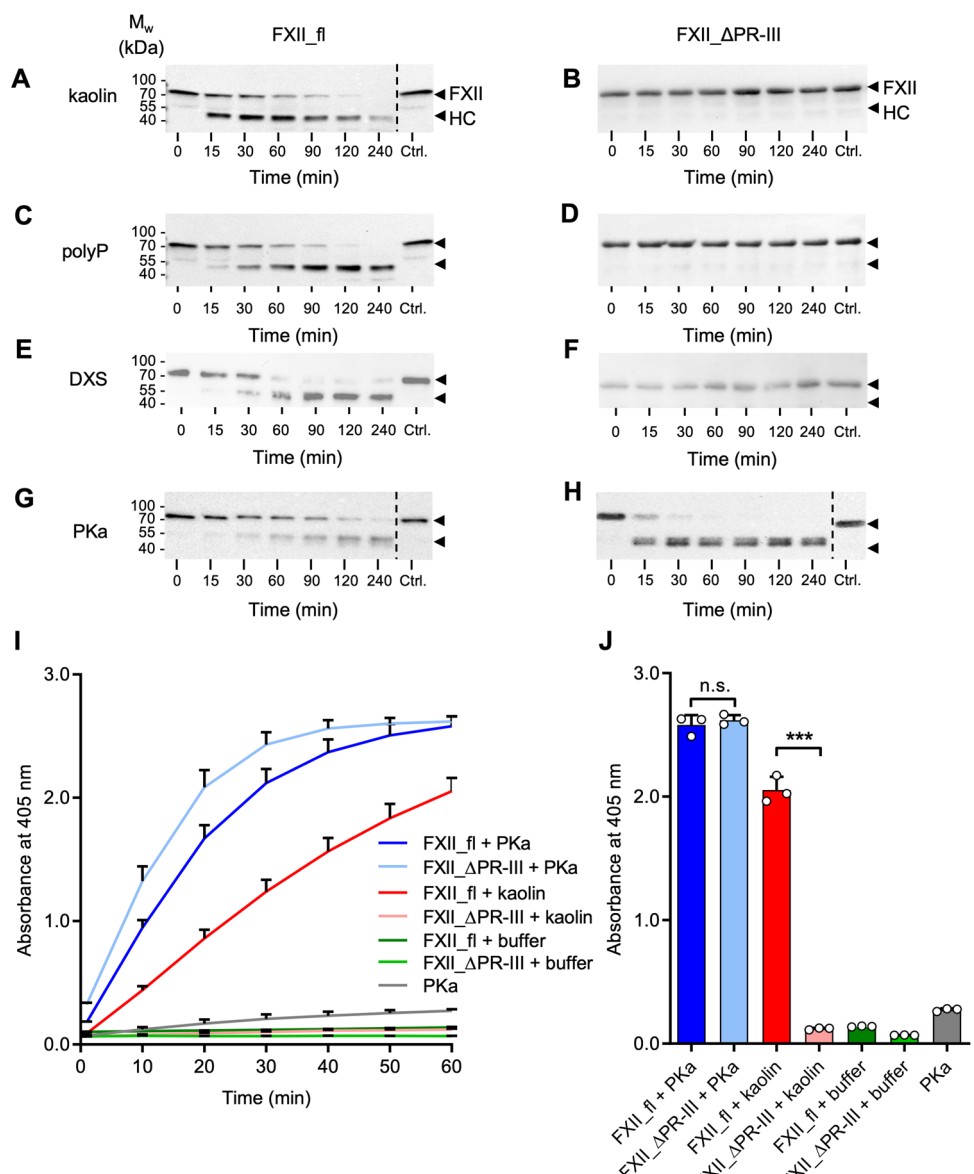

**Fig. 3 FXII_ΔPR-III is defective in contact activation but susceptible for fluid phase activation. A–H** Recombinant FXII_fl or FXII_ΔPR-III were incubated with kaolin (100 µg/mL; **A**, **B**), short-chain soluble polyP (500 µg/mL; **C**, **D** dextran sulfate (DXS, 1 µg/mL; **E**, **F**) or plasma kallikrein (PKa, 30 nM; **G**, **H**) for 0, 15, 30, 60, 90, 120, and 240 min at 37 °C. Samples were analyzed under reducing conditions by western blotting using polyclonal anti-FXII antibodies. Control (Ctrl): Recombinant FXII_fl or FXII_ΔPR-III incubated with buffer only for 240 min. FXII: FXII zymogen, HC: FXIIa heavy chain. Representative images of $n = 3$, are shown. **I, J** FXIIa formation was measured by the conversion of the chromogenic substrate S2302 at 405 nm. To ensure that formed FXIIa cleaved the substrate, PKa-incubated samples were treated with aprotinin (100 KIU/mL) to neutralize PKa before S2302 conversion measurement was started. $n = 3$ samples, experiments independently performed 3 times. n.s.: non-significant ($P > 0.05$), ***: $P < 0.001$, by unpaired two-tailed Student's $t$ test. Data are mean values ± SD.

$10.7 \pm 2.5$ min, respectively. In contrast, $F12^{-/-}$ mice reconstituted with FXII_ΔPR-III survived the collagen/epinephrine challenge (4/5) with survival times of $32.5 \pm 5.6$ min. To confirm PE formation, lung perfusion was measured in all challenged mice using intravenous administration of Evans blue dye (Fig. 5C, upper panel). Perfused lung areas turned blue, whereas occluded parts retained their natural pink colour. Collagen/epinephrine challenge resulted in almost complete vascular thrombotic occlusion in WT and FXII_fl-reconstituted $F12^{-/-}$ mice, as visualized by the disturbed perfusion of the dye. In contrast, lungs of FXII_ΔPR-III expressing $F12^{-/-}$ mice presented with uniform distribution of the dye similar to $F12^{-/-}$ controls, consistent with preserved vessel perfusion. Furthermore, $F12^{-/-}$ mice reconstituted with FXII_ΔPR-III but not FXII_fl were protected from i.v. infused kaolin and resultant lethal

PE (Supplementary Fig. 4A). Consistent with these thrombosis studies, addition of FXII_fl normalized the prolonged kaolin-initiated clotting time of $F12^{-/-}$ mouse plasma to WT levels ex vivo ($32 \pm 4$ *vs.* $29 \pm 3$ s). In contrast, addition of FXII_ΔPR-III was unable to normalize defective clotting of $F12^{-/-}$ mouse plasma ($92 \pm 18$ *vs.* $96 \pm 19$ s). In contrast to the critical role of PR-III in thrombus formation, infusion of MBP-PR-III had no impact on bleeding time or blood loss both in WT and $F12^{-/-}$ mice (Supplementary Fig. 4B, C). Activated platelets initiate coagulation in a FXII-dependent manner[33]. To confirm the role of PR-III in platelet-driven contact activation, we compared thrombin formation triggered by procoagulant platelets in FXII-deficient plasma reconstituted with FXII_fl or FXII_ΔPR-III. Activated platelets ($25 \mu$M ionomycin[25]) initiated coagulation in FXII-deficient plasma

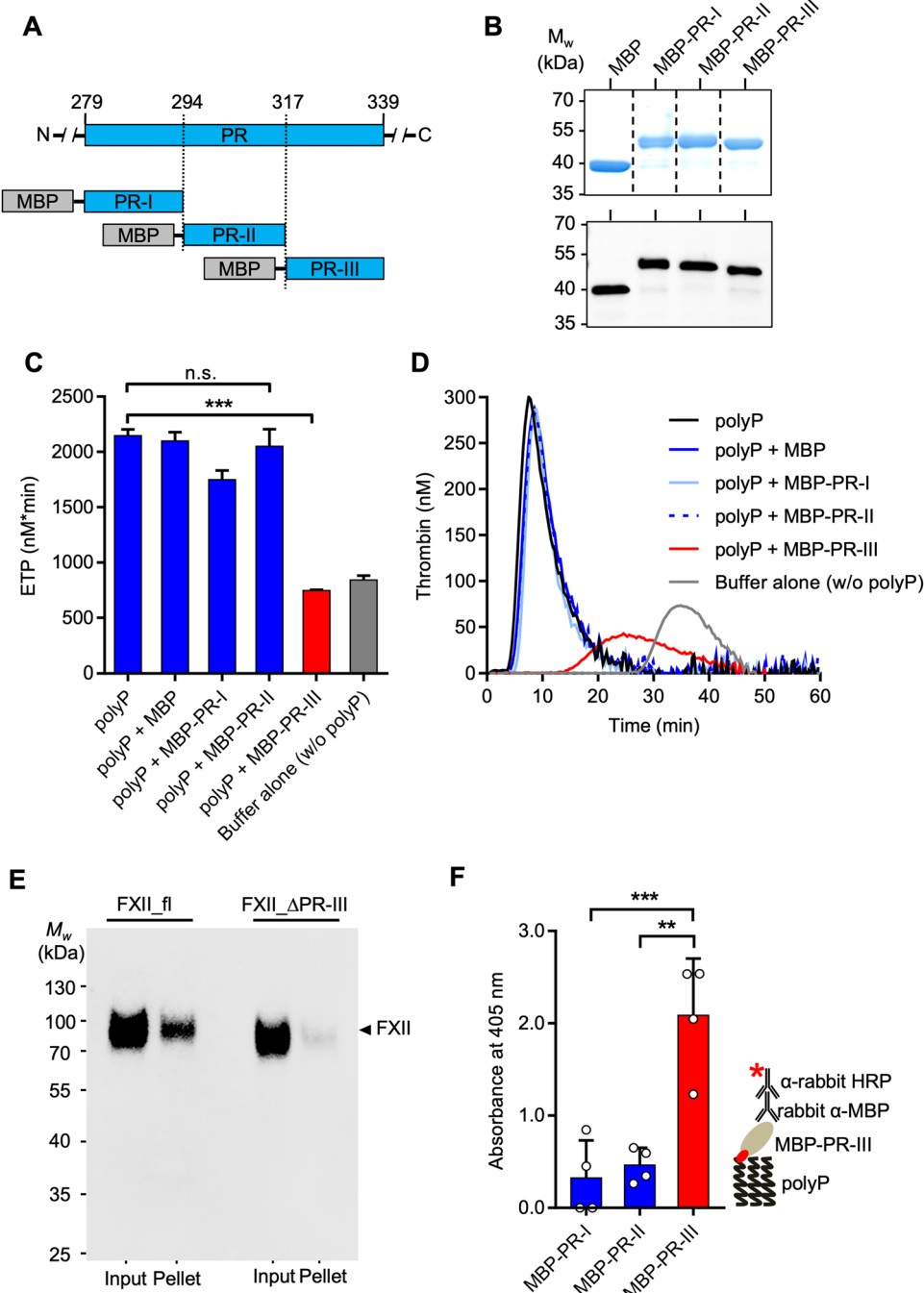

**Fig. 4 The PR-III sequence mediates FXII contact activation. A** Schematic overview of maltose binding protein (MBP) fused to PR-I, PR-II or PR-III forming MBP-PR-I, MBP-PR-II and MBP-PR-III, respectively. **B** Analysis of purified MBP, MBP-PR-I, MBP-PR-II and MBP-PR-III using Coomassie blue-stained reducing SDS-PAGE (top panel, 2 µg protein/lane) or western blotting with an anti-MBP antibody (bottom panel, 40 ng protein/lane). n = 2. **C** ETP of normal PPP stimulated with long-chain polyP (5 µg/mL), incubated with MBP, MBP-PR-I, MBP-PR-II or MBP-PR-III (600 nM), or buffer alone (no polyP), n = 3 run in triplicate each. n.s.: non-significant, ***: P < 0.001 vs. polyP by one-way ANOVA and Dunnett´s multiple comparison test. **D** Representative real time thrombin generation curves of **C**. **E** Pulldown of FXII_fl and FXII_ΔPR-III with kaolin. Input and washed pellet were probed by anti-FXII antibodies in western blotting. One blot of n = 2. **F** Binding of MBP-PR-I, MBP-PR-II, and MBP-PR-III to immobilized polyP. Bound proteins were quantified photometrically using anti-MBP and HRP-coupled detection antibodies followed by substrate reaction. Inset: Schematic representation of the ELISA assay. n = 4. ***: P = 0.0006, **: P = 0.001 by one-way ANOVA and Dunnett´s multiple comparison test. Mean values ± SD. HRP: horseradish peroxidase.

supplemented with FXII_fl, however were inactive in triggering thrombin generation in the presence of FXII_ΔPR-III (Fig. 5D).

**Polyclonal antibodies against PR-III initiate FXII activation.** We produced polyclonal antibodies against the 14-residue peptide Gly320-Thr333, located in the PR-III core sequence (pAB-

αGAL14; Fig. 6A, upper panel). Immunopurified pAB-αGAL14 recognized endogenous FXII in normal PPP, purified FXII_pd, and recombinant FXII_fl but not FXII_ΔPR-III (Supplementary Fig. 5A). As PR-III mediated surface binding activates FXII, we hypothesized that pAB-αGAL14 binding to FXII would also cause its activation. In contrast to insoluble particles with a poorly

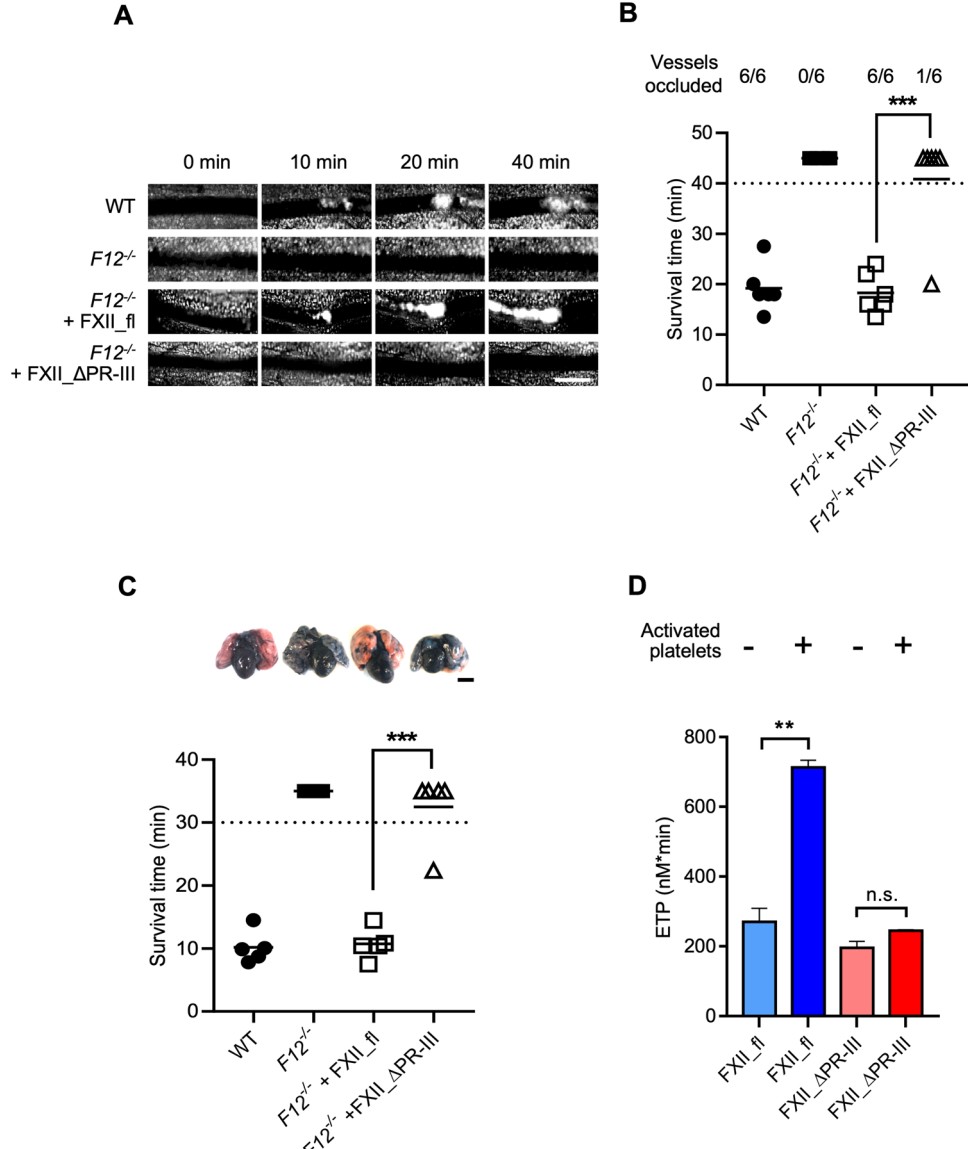

**Fig. 5 Defective thrombosis in FXII_ΔPR-III reconstituted *F12*$^{-/-}$ mice. A** Thrombus formation in vivo was monitored by intravital microscopy of mesenteric arterioles following topical application of 20% FeCl$_3$ in wild type (WT, top lane) and *F12*$^{-/-}$ mice, reconstituted with buffer (second lane), FXII_fl (third lane) or FXII_ΔPR-III (bottom lane). Representative images of $n = 6$ experiments. Scale bar equals 50 μm. **B** Time to complete occlusion after injury. Each symbol represents one monitored arteriole. The number of occluded vessels during the 40 min observation period is given at the top. $n = 6$ mice. **C** Pulmonary embolism induced by intravenous infusion of collagen/epinephrine. The survival time of WT, *F12*$^{-/-}$ or *F12*$^{-/-}$ mice with hydrodynamic tail vein injection-mediated transgene expression of FXII_fl or FXII_ΔPR-III was monitored. Mortality was assessed in each group of mice and animals alive 30 min after the challenge, were considered survivors; Top panel: Collagen/epinephrine-challenged mice were intravenously infused with Evans blue shortly after the onset of respiratory arrest and while the heart was still beating or for survivors, at 30 min post-challenge. Lungs were harvested and perfusion was determined. Occluded lung parts remain their natural pinkish colour. The bar represents 5 mm. $n = 5$ mice. In **B**, **C**: ***: $P < 0.001$, by unpaired two-tailed Student's t test. Each symbol represents one animal. **D** ETP triggered by activated platelets (25 μM ionomycin, $5 \times 10^7$ platelets/mL) or buffer in FXII_fl or FXII_ΔPR-III reconstituted FXII-deficient plasma. $n = 4$ PRP samples. n.s.: non-significant ($P > 0.05$), **: $P = 0.005$, by unpaired two-tailed Student's t test. Columns give means ± SD.

defined activity in inducing FXII contact activation, surface-mimicking pAB-αGAL14 antibodies would allow for regulated stoichiometric activation of FXII within a homogenous solution. Indeed, treatment of FXII_pd with pAB-αGAL14 (final concentration 750 nM; 5:1 molar ratio of antibody:FXII in the reaction mixture) potently triggered FXIIa formation (Fig. 6B). A control antibody against cytoskeleton vasodilator-stimulated phosphoprotein (VASP, pAb-αControl; 750 nM) was inactive towards generating FXIIa activity. Using recombinant N-terminal FXII deletion mutants, others have shown that EGF-I domain is

indispensable for surface binding and likely mediates contact activation[34]. To validate a potential function of EGF-I for contact activation we produced polyclonal antibodies against the EGF-I peptide Ser83-Cys102 (pAB-αSPC18; Fig. 6A, lower panel) that recognized FXII_pd, FXII in normal PPP, and FXII_fl but not FXII_ΔEGF-I (Supplementary Fig. 5B). pAB-αSPC18 antibodies up to 750 nM (5:1 molar ratio), failed to induce FXII contact activation (Fig. 6C). In contrast, pAB-αGAL14 initiated coagulation in normal PPP in a dose-dependent manner (ETP of 2090 nM*min at 3:1 molar excess over plasma FXII), while control

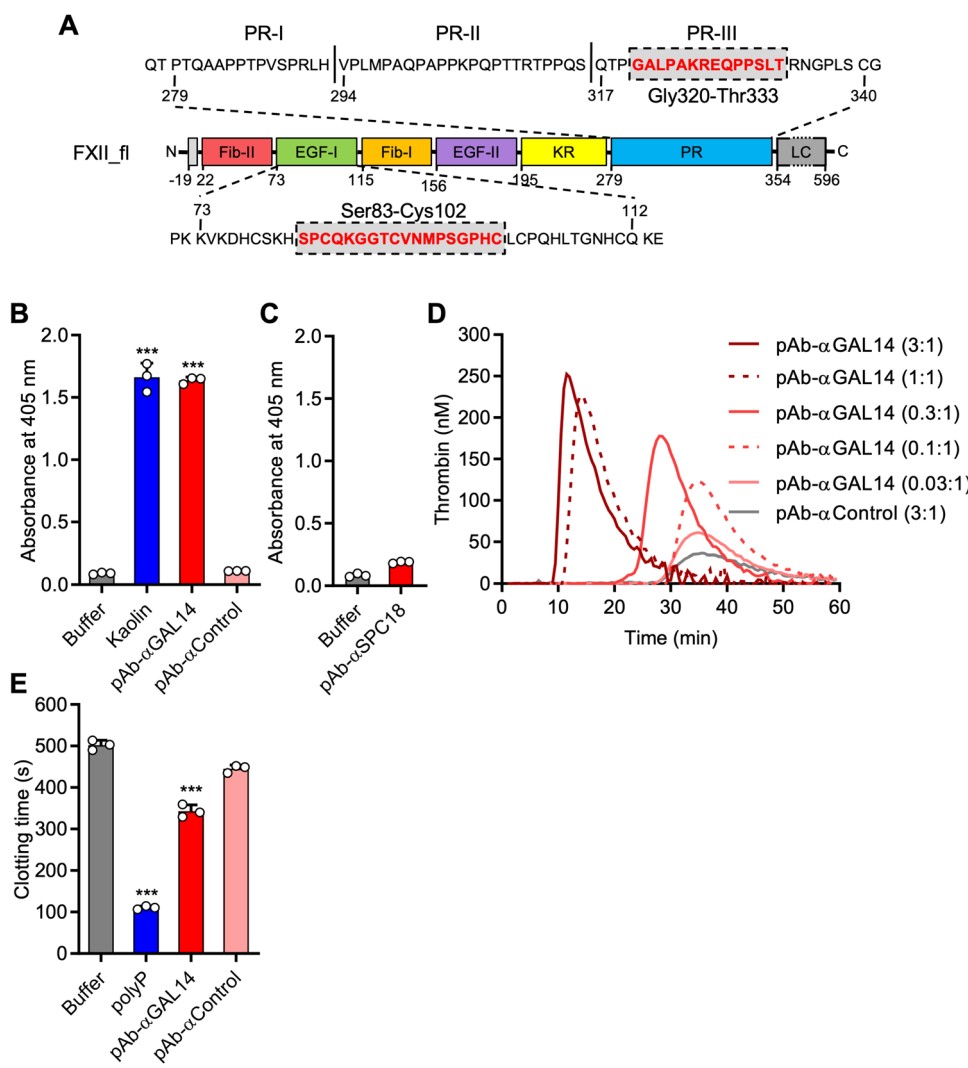

**Fig. 6 Polyclonal antibodies against PR-III activate FXII. A** Schematic structure of human full-length FXII. Top panel: PR-I, PR-II, and PR-III amino acid sequences are indicated within the PR domain; Bottom panel: sequence of the EGF-I domain. The peptide sequences used to raise the polyclonal antibodies pAB-αGAL14 (Gly320-Thr333) and pAB-αSPC18 (Ser83-Cys102) are highlighted in red. Numbers indicate amino acid residues. **B** FXII_pd was incubated with buffer, kaolin (10 μg/mL), pAB-αGAL14 (1 μM) or a control polyclonal antibody (pAB-αControl, 1 μM). FXIIa formation was measured using S2302. **C** FXII_pd was incubated with buffer or pAB-αSPC18 (1 μM) and FXIIa formation was measured using S2302. **D** Real-time thrombin generation in normal PPP activated with pAB-αGAL14 or pAB-αControl. Molar ratio antibody:FXII are relative to endogenous FXII levels in normal PPP (375 nM). Representative thrombin generation curves are shown. (E) Clotting times of recalcified normal PPP incubated with buffer, long-chain polyP (25 μg/mL), pAB-αGAL14 (2 μM) and/or pAB-αControl (2 μM). In **B**, **E**: ***: $P < 0.001$ vs. buffer by one-way ANOVA and Dunnett´s multiple comparison test. Columns give means ± SD. For **B**–**E** three independent experiments of $n = 3$ each were performed.

antibodies did not trigger thrombin formation even at the highest concentration tested (pAB-αControl, ETP 100 ± 10 nM*min at a 3:1 ratio; Fig. 6D). To study the capacity of pAB-αGAL14 to activate clotting, we incubated normal PPP with buffer, pAb-αControl or pAB-αGAL14. pAB-αGAL14 treatment shortened clotting times from 502 ± 12 s to 343 ± 15 s (10:1 molar ratio, $P < 0.001$ vs. buffer) while pAb-αControl was inactive (445 ± 9 s, $P > 0.05$; Fig. 6E).

**Monoclonal antibodies initiate controlled FXII activation.** A high degree of precision, reproducibility and standardization are crucial features for diagnostic coagulation assays. To develop an assay of antibody-driven FXII activation that can be scaled for clinical use, we generated monoclonal anti-Gly320-Thr333 antibodies designated mAb-αPR-III_9, mAb-αPR-III_27, mAb-αPR-III_37, and mAb-αPR-III_43. In an ELISA, mAb-αPR-III_27, mAb-αPR-III_37, and mAb-αPR-III_43, but not mAb-αPR-III_9,

cross-reacted with immobilized Gly320-Thr333 peptide (Supplementary Fig. 6A) and FXII_pd (Fig. 7A) and were selected for further analysis. mAb-αPR-III_27 and mAb-αPR-III_37 but not mAb-αPR-III_43 recognized endogenous FXII in plasma samples by immunoblotting (Fig. 7B). Incubating FXII_pd with mAb-αPR-III_27 (5:1 molar ratio) initiated FXIIa activity over background levels (≈1.5-fold), however, mAb-αPR-III_37 was >4-fold more potent in inducing protease formation from FXII zymogen (Fig. 7C). The most potent FXII activator mAb-αPR-III_37 was selected for further studies. To analyse the mechanism and consequences of antibody-triggered FXIIa formation, we stimulated normal PPP with mAb-αPR-III_37 (5:1 molar ratio). The surface-mimicking antibody increased FXIIa formation >5-fold over buffer control (Fig. 7D). Antibody-triggered conversion of the S2302 substrate was completely dependent on FXII and was absent in FXII-deficient plasma (Fig. 7E). Enzymatic activity induced by mAb-αPR-III_37 was blocked by FXIIa inhibitor corn

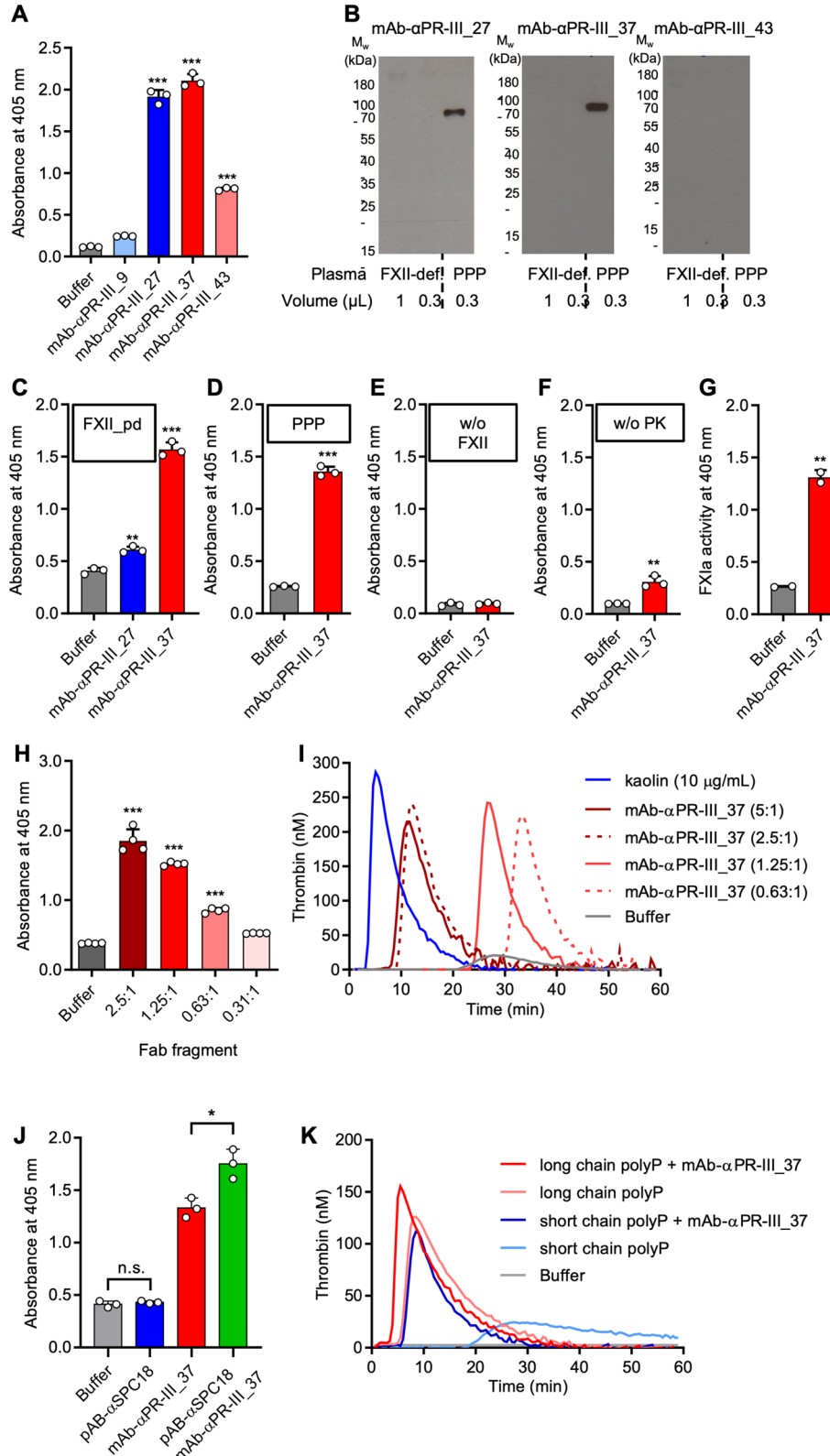

trypsin inhibitor (2 μM) to background levels. Supporting the specificity of mAb-αPR-III_37 in activating FXII in plasma, the magnitude of antibody-stimulated FXIIa correlated with plasma FXII zymogen levels (Supplementary Fig. 6B). Additionally, western blotting revealed that mAb-αPR-III_37 initiated FXII zymogen cleavage in plasma dose-dependently. In PK-deficient plasma, fluid phase activation and reciprocal FXII(a)/PK(a)

activation are defective. In PK-deficient plasma, mAb-αPR-III_37 increased FXIIa formation ~3-fold over buffer control (Fig. 7F). These data indicate that ~60% and 40% of formed FXIIa originated from mAb-αPR-III_37-driven contact activation and PKa cleavage, respectively. Addition of mAb-αPR-III_37 to normal PPP produced FXIIa and led to the generation of activated FXI (FXIa), detected by proteolysis of the FXIa-specific chromogenic

**Fig. 7 Surface-mimicking monoclonal anti-PR-III antibodies initiate FXII contact activation. A** Binding of purified monoclonal anti-PR-III antibodies mAb-αPR-III_9, mAb-αPR-III_27, mAb-αPR-III_37 or mAb-αPR-III_43 (0.3 μM each) to immobilized FXII_pd in an ELISA assay. FXII-bound antibody was measured photometrically at 405 nm following substrate reaction. n = 3, experiment performed 3 times. **B** FXII-deficient plasma samples (FXII-def; 1 μL in lane 1 and 0.3 μL in lane 2), or normal platelet poor plasma (PPP; 0.3 μL in lane 3) were probed with mAb-αPR-III_27, mAb-αPR-III_37 or mAb-αPR-III_43 in western blot analyses, n = 2. **C** FXII_pd was incubated with buffer, mAb-αPR-III_27 or mAb-αPR-III_37 (750 nM each) and FXIIa formation was monitored by S2302 conversion. **D–F** Normal PPP (**D**), FXII-deficient plasma (**E**) or plasma kallikrein (PK)-deficient plasma (**F**) was incubated with buffer or mAb-αPR-III_37 (750 nM) and S2302 substrate cleavage was measured at 180 min. **G** Normal PPP was incubated with buffer or mAb-αPR-III_37 (750 nM) for 60 min and FXIa formation was measured in a chromogenic assay using the S2366 substrate. **H** Normal PPP was activated with decreasing concentrations of purified mAb-αPR-III_37 Fab fragments (molar ratio of 2.5:1 to 0.31:1 of Fab fragment:endogenous normal PPP FXII) and FXIIa formation was measured using the S2302 chromogenic assay. **C–H**: n = 3 of 3 independent experiments. **I** Real-time thrombin generation in normal PPP stimulated with mAb-αPR-III_37 or kaolin (10 μg/mL). Antibody concentrations are given relative to endogenous normal PPP FXII (375 nM). **J** Normal PPP was incubated with pAB-αSPC18 or mAb-αPR-III_37 alone (each 750 nM), or a mixture of both antibodies. FXIIa formation was measured by S2302 conversion. n.s.: non-significant, n = 3; one dataset of two is shown. **K** Real-time thrombin generation in normal PPP stimulated with long- or short-chain polyP (10 μg/mL each), in the presence or absence of mAb-αPR-III_37 (125 nM, molar ratio of 1:1) or buffer. Representative thrombin generation curves of n = 4, are shown. In **A**, **C**, and **H** **: P = 0.0034, ***: P < 0.001 vs. buffer by one-way ANOVA and Dunnett's multiple comparison test. For **D–G** and **J** *: P = 0.0126, **: P < 0.01, ***: P < 0.001 by unpaired two-tailed Student's t test. All columns give means ± SD.

substrate, S2366 (4-fold over control; Fig. 7G). Comparable to intact mAb-αPR-III_37, purified Fab-fragments from mAb-αPR-III_37 antibody triggered FXIIa formation (Fig. 7H). Real-time thrombin generation assays indicated that mAb-αPR-III_37 initiated normal PPP coagulation in an antibody dose-dependent manner (Fig. 7I). As shown in Fig. 6C, addition of pAB-αSPC18 alone did not induce FXII activation. However, pAB-αSPC18 significantly amplified mAb-αPR-III_37 activity towards FXII contact activation (Fig. 7J). Consistently, the contact activating potential of mAb-αPR-III_37 was >4-fold increased by miniscule amounts of short and long-chain polyP or DXS that alone were unable to induce FXII contact activation (Supplementary Fig. 6C–E). Long-chain polyP is a more potent FXII contact activator compared to the short-chain polymer[35] and equal amounts of long-chain polyP (10 μg/mL) initiated coagulation more potently compared to the short-chain polymer (Fig. 7K). Co-application of long-chain polyP (1 μg/mL) and mAb-αPR-III_37 (125 nM, molar ratio of 1:1) shortened clotting time of normal PPP to 78 ± 2 s, exceeding levels of polyP alone (120 ± 2 s).

**mAb-αPR-III_37 as a tool for coagulation diagnostics.** Defined concentrations of mAb-αPR-III_37 activate FXII in solution in a regulated manner, leading to plasma coagulation. Insoluble particles with ill-defined surfaces, such as silica and ellagic acid are used as reagents to trigger FXII contact activation in diagnostic aPTT assays. Due to their surface-dependent FXII-activating potential, heterogeneity among contact activators and even lot-to-lot differences have prevented standardization of the aPTT assay. When we compared multiple particulate aPTT reagents from various manufacturers for their degree of FXIIa generation, we found a considerable degree of variation (4- to 10-fold increase of FXIIa formation, as compared to buffer control; Fig. 8A). A defined concentration of mAb-αPR-III_37 (5:1 molar ratio) generated reproducible FXIIa activity and thrombin formation in solution and in a homogenous phase. Based on our studies, we propose that mAb-αPR-III_37 can be used for standardization of contact activation elicited by insoluble particulate aPTT reagents.

Classically, the aPTT is used as a screening test for (intrinsic) coagulation pathway disturbances. Deficiency in coagulation factors VIII or IX (FVIII and FIX) underlies the bleeding disorders haemophilia A and B, respectively. Therefore, it is imperative that a screening diagnostic coagulation assay can accurately capture the presence of coagulopathy. Plasma samples containing 0–5% FVIII or FIX levels of normal PPP, respectively, were incubated with mAb-αPR-III_37 (1:1 molar ratio) and downstream formation of activated factor X (FXa) was monitored

using the chromogenic substrate S2222. FXa signal correlated with plasma levels of FVIII or FIX, thus allowing for accurate determination of FVIII or FIX levels (Fig. 8B, C).

Inhibition of FXI has emerged as a promising strategy for novel anticoagulants[11]. In contrast to direct thrombin or factor X inhibitors (DOACs), thromboprotection requires largely reduced factor levels and is conferred when FXI activity is reduced to approximately <25% of normal. A significant drawback of FXI inhibitors is the lack of sensitive coagulation assays that can precisely measure FXIa activity over a broad range of FXI levels. Here, we utilized plasma samples containing 1–100% FXI and these were incubated with mAb-αPR-III_37 (5:1 molar ratio) or buffer control, upon which and FXIa formation was monitored (Fig. 8D). mAb-αPR-III_37 precisely and reproducibly detected FXI plasma levels as low as ~1%. In contrast, various commercial aPTT reagents exhibited low sensitivity at measuring FXI levels in the predefined, low therapeutic range (Fig. 8E).

## Discussion

FXII contact activation is critically involved in blood clotting following exposure to a broad variety of synthetic and physiologic surfaces[36], and this reaction provides the underlying principle for one of the most commonly measured diagnostic clotting tests, the aPTT assay[13,37]. However, the exact mechanism of FXII contact activation remains unresolved. Using three complementary strategies based on FXII deletion mutants, contact activation competitors and FXII-activating antibodies, we show that a continuous segment of 23 amino acid residues located in the C-terminal part of the PR domain (PR-III, Gln317-Ser339) is essential for FXII contact activation in plasma. "Contact" of FXII to surface-mimicking mAb-αPR-III_37 antibody produces FXIIa activity in solution without a surface. mAb-αPR-III_37 allows for the development of novel coagulation assays beyond current applications of aPTT diagnostic tests, such as precise measurement of intrinsic coagulation factor activities over a large range and standardization of particulate aPTT reagents.

In addition to activation following contact, several proteases including PKa, plasmin, FXIa, thrombin, trypsin and an array of pathogen-derived enzymes, can proteolytically activate FXII zymogen in plasma [reviewed in[2]]. Similar to deficiency of full-length FXII[30], deletion of FXII PR-III abolished occlusive thrombus formation in the ferric chloride and collagen/epinephrine thrombosis models (Fig. 5), suggesting use of PR-III blocking agents for anticoagulation. These data indicate that contact activation mediated FXIIa formation plays a distinct and significant role in thrombosis. FXII has limited proteolytic activity in its zymogen form[38] providing a rationale for the minute

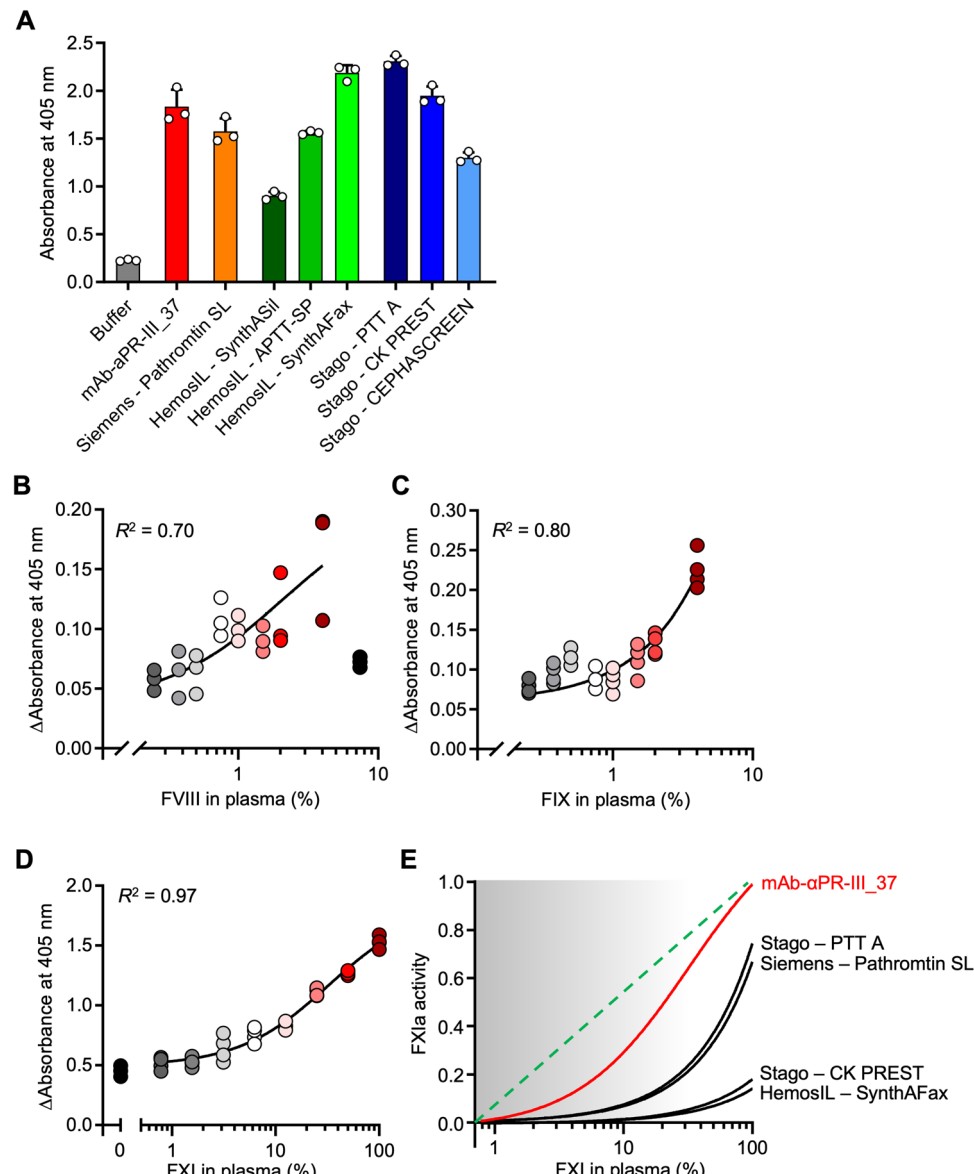

**Fig. 8 mAb-αPR-III_37 as a tool for coagulation diagnostics. A** Standardization of aPTT reagents: Normal PPP was incubated with buffer, mAb-αPR-III_37 (750 nM, molar ratio antibody:plasma FXII of 5:1) or commercially available aPTT reagents (diluted 1:200). FXIIa formation was measured by S2302 cleavage. $n = 3$ samples, experiments independently performed 8 times. Columns give means ± SD. **B**, **C** Analysis of FVIII and FIX deficiency: Normal PPP was mixed with FVIII-deficient (**B**) or FIX-deficient (**C**) plasma to establish the precise FVIII and FIX levels in each reaction. Samples were incubated with mAb-αPR-III_37 (37.5 nM, molar ratio of 1:1 with plasma FXII) or buffer in the presence of CaCl₂, phospholipids and H-Gly-Pro-Arg-Pro-OH acetate salt for 30 min. FXa formation in mAb-αPR-III_37-treated samples was measured photometrically using S2222 conversion for 180 min and corrected for the absorbance of buffer-stimulated samples. **B**, **C** shows $n = 3$ samples, experiments independently performed 5 times. **D** FXI analysis: Normal PPP and FXI-deficient plasma were mixed to obtain defined FXI plasma concentrations. Samples were incubated with mAb-αPR-III_37 (750 nM, molar ratio antibody:plasma FXII of 5:1) or buffer for 180 min. FXIa formation in antibody stimulated samples was measured using the S2366 substrate and was blotted following correction for buffer, background levels. **E** mAb-αPR-III_37 but not particulate aPTT agents accurately measures FXI plasma levels: Normal PPP and FXI-deficient plasma were mixed to obtain defined FXI plasma concentrations. Plasmas were incubated with mAb-αPR-III_37 (5:1 molar ratio) or various aPTT reagents. FXIa formation was measured using the S2366 substrate and were blotted following correction for FXI-deficient plasma background levels. $R^2$-values were calculated by a non-linear regression curve (agonist *vs.* response, 3 parameters). **D**, **E** shows $n = 4$, experiments independently performed 5 times.

procoagulant activity of contact activation-deficient FXII mutants (Fig. 2A).

Previously, epitope mapping of inhibitory FXII antibodies interfering with contact activation identified several putative sites on FXII involved in surface-binding and contact activation. Monoclonal antibodies P5-2-1 and B7C9 interfered with surface-triggered FXIIa formation in plasma[39] and epitopes of these antibodies were originally mapped to residues Ile1-His28 and

Thr134-Arg153 in the Fib-II and Fib-I domains of the heavy chain, respectively[40]. A follow-up study revealed that P5-2-1 and B7C9 competed with each other for FXII binding and they were both mapped to epitope Ile1-His28[41]. However, a FXII deletion mutant lacking residues Pro3-Val19 was readily activated following exposure to the contact activator DXS, challenging the notion that the extreme N-terminus of the FXII heavy chain is involved in surface-binding of FXII. In fact, the contact-activation

potential of the Pro3-Val19 mutant exceeded that of normal FXII by 4-fold, despite the fact that B7C9 failed to bind to the ΔPro3-Val19 mutant[42]. Anti-FXII heavy chain antibody KOK5 inhibited FXII contact activation induced by kaolin[43]. In silico modelling suggested that KOK5 cross-reacted with a discontinuous epitope in the Fib-II domain, involving residues Phe30-Gln33, His40-Arg47, and Thr57-Phe60. Accordingly, a synthetic peptide spanning residues Tyr39-Arg47 interfered with FXII contact activation, supporting the role of the Fib-II domain in FXII contact activation. However, our FXII mutants lacking the Fib-II domain, alone (FXII_ΔFib-II) or in combination with EGF-I (FXII_ΔFib-II/EGF-I; Fig. 1), were susceptible to contact activation triggered by kaolin or polyP (Fig. 2), indicating that the Fib-II and EGF-I domains are not crucial for FXII contact activation. The reason for these discrepant results using antibodies and FXII mutants is not entirely clear but may reflect steric interference of bulky antibodies with FXII surface binding.

The F1 antibody activates FXII in plasma[44] and its epitope was mapped to the KR domain[45]. The mechanism underlying F1-facilitated FXIIa formation remains to be shown, however it likely involves induction of conformational changes, stabilization of intermediate forms occurring during FXII activation, or increased PKa-mediated zymogen activation[38,46]. PKa and plasmin proteolytically activate FXII zymogen in solution in a surface-independent manner. The current study shows that deficiency in PR-III eliminates FXII from surface-induced activation (Fig. 2), however concomitantly facilitates FXII activation in solution by PKa (Fig. 3G, H), thrombin, and plasmin (Supplementary Fig. 2G, H).

Pull-down studies using recombinant FXII N-terminal deletion mutants suggested that FXII_fl and FXII_ΔFib-II, but not FXII_ΔFib-II/EGF-I, bind to kaolin[34]. The data indicated that the EGF-I domain contains a surface-binding site. However, in the current study, FXII deletion of the EGF-I domain did not interfere with contact activation following exposure to kaolin or polyP (Fig. 2), showing that EGF-I is dispensable for contact activation. Consistently, and in contrast to antibodies against PR-III (Figs. 6 and 7), antibodies against the core sequence of EGF-I did not induce FXII activation (Fig. 6C).

Our current data and previously published work propose a two-step mechanism for FXII contact activation. Initially, the interaction between FXII and a surface commences by binding of FXII to EGF-I and/or possibly the Fib-II domain. This interaction induces a conformational shift that unshields and exposes the contact activation site located within the PR-III region. In line with a possible inhibitory-function of the N-terminal heavy chain, a FXII_ΔFib-II/EGF-I mutant[34] and a Trp268Arg substitution in the KR domain[46], largely increased the susceptibility for contact activation. In our hands, it was challenging to produce the FXII_ΔFib-II/EGF-I mutant in a purely zymogen form (Supplementary Fig. 7), while others reported that the variant activated "spontaneously" with about 20% of the protein readily activated in the supernatant of transfected cells[34]. A recombinant FXII mutant spanning the EGF-II, KR and PR domains and the light chain (rFXII-U-like) binds to surfaces and undergoes contact activation[47], demonstrating that the N-terminus of the heavy chain is dispensable for zymogen contact activation. While the Fib-II/EGF-I domains appear to regulate the activation threshold of FXII, it is the PR-III region that is indispensable for activation. PR-III contains another surface binding site (Fig. 4E, F) and surface binding to PR-III is sufficient and a prerequisite for FXII contact activation (Figs. 2A, B and 7J). Consistently, MBP-PR-III competes with surface-driven FXII zymogen activation (Fig. 4C, D) and a FXII mutant (rFXII.lpc) lacking the 319 N-terminal amino acids of the heavy chain but containing the PR-III region, bound to kaolin and was activated by a surface[48]. Together, the data are consistent with a two-step model of FXII contact

activation comprised of (i) zymogen priming binding to the Fib-II and/or EGF-I domains that exposes PR-III and (ii) facilitates binding by a surface to PR-III that ultimately leads to FXII activation. Antibodies that specifically target the priming and activation site allowed validation of this mechanism. While antibodies against the EGF-I domain alone failed to trigger FXII activation (Fig. 6C), they amplified the FXIIa-forming activity of mAb-αPR-III_37 (Fig. 7J).

The proposed mechanism of FXII activation provides insight into the clinical manifestations of naturally occurring FXII variants. FXII-HAE arises by a variant of FXII whereby Thr309 is substituted with a lysine or arginine (FXII-Lys/Arg309;[49,50]). FXII Lys/Arg309 variants exhibit defective glycosylation which leads to a FXII structure that is more readily activated in the presence of a surface[51,52]. Recently N-glycosylation at Asn230 was identified[53], offering an explanation for the low apparent molecular mass of FXII_ΔKR mutant lacking that specific site. HEK239 produced FXII_fl and FXII_ΔFib-I migrated with highly similar apparent molecular weights, suggesting that the SDS-PAGE system is not sensitive enough to capture small differences in high molecular weight glycoproteins. The minor difference in apparent molecular weight of FXII_fl and FXII_ΔFib-I was better visible in western blot analyses of HepG2 and CHO expressed mutants that used identical DNA vector constructs (Supplementary Fig. 1A, B). In addition to the FXII Thr309 missense mutations, HAE has been associated with either a F12 exon 9 deletion (c.971_1018 + 24del72*)[54] or a duplication of 18 base pairs (c.892_909dup) that lead to loss of amino acids Lys305-Ala321 or repetition of amino acids (Pro298-Pro303)[55], respectively. Although the precise mechanisms by which these mutations lead to HAE remains to be established, it raises the possibility that their proximity to the PR-III region can drive an exuberant FXII fluid phase activation (Fig. 3G, H; Supplementary Fig. 2G, H) and bradykinin formation, which results in tissue oedema in these patients.

The complex glycosylation pattern within the PR region makes it difficult to draw definite structural conclusions for deletions in this domain. Furthermore, proline-rich regions mediate binding by preferentially adopting a flexible polyproline type II helix structure that facilitates transient interactions (reviewed in[56]), together providing a rationale for the enhanced sensitivity to contact activation, in the absence of PR-II/PR-III as compared to PR-III deletion alone (Fig. 2A).

Activation of FXII by a negatively charged surface underlies the aPTT assay, the most frequently used screening coagulation test[57]. Despite its global use, the diagnostic accuracy and sensitivity of the aPTT assay largely depends on the type of particulate reagent used[57]. To overcome this limitation, we propose that the use of mAb-αPR-III_37 allows for standardization and improved sensitivity across aPTT reagents (Fig. 8A). Classical aPTT assays are not precise for monitoring unfractionated heparin (UFH) and DOACs (with the exception of thrombin inhibitor dabigatran etexilate), and plasma drug levels do not correlate with the degree of aPTT prolongation[58]. Particulate aPTT is also insensitive at measuring low FXI levels (Fig. 8E) and correlation between FXI levels as determined by aPTT reagents and incidence of bleeding is weak[59,60]. The potential diagnostic use of mAb-αPR-III_37-based assays to identify FXI deficient individuals at risk of bleeding is worth pursuing in future studies. Moreover, our data show that other intrinsic coagulation factors (FVIII and FIX) can be accurately measured at levels as low as 0–5% (Fig. 8B, C). Furthermore, aPTT assays triggered by surfaces were also not sensitive at assessing UFH levels (Supplementary Fig. 8), which can pose significant bleeding risk for treated patients. mAb-αPR-III_37 antibody-based assays may provide novel opportunities for diagnostic coagulation laboratories to accurately assess UFH and DOAC anticoagulant effects, and in the case of patients receiving

UFH, to enable worldwide standardized recommendations for the aPTT goal range. However, the diagnostic utility of mAb-αPR-III_37 requires further clinical studies.

In conclusion, the current study provides key structural details into the mechanism of FXII contact activation and introduces novel tools for improved coagulation diagnostics. These advances are expected to have far reaching clinical applications and considerably improve patient care.

## Methods

**Reagents**. Plasma derived FXII, PKa, and PK were obtained from Molecular Innovations. FXII-, FXI-, FVIII-, FIX-, and PK-deficient plasmas from individuals with congenital factor deficiency were obtained from George King Bio-medical and had no detectable antigen level by western blotting and did not exhibit coagulant activity for each respective protein. FXII-deficient plasma samples were from three different lots. Aprotinin was obtained from Bayer. Synthetic soluble polyP (60- to 100-unit chain length) and insoluble polyP (100- to 400-unit chain length) were donated by Dr. Thomas Staffel, ICL Pharmaceutical. PolyP was isolated from *E. coli* by phenol-chloroform extractions and Dowdex 50 W ion-exchange resin[25]. DXS 500.000 was from Sigma-Aldrich. Plasmin and thrombin were purchased from Hyphen BioMed and Sigma-Aldrich, respectively. Dabigatran was from Cayman Chemical and FXa from Haematologic Technologies. S-2302 (H-D-prolyl-L-phenylalanyl-L-arginine-p-nitroaniline dihydrochloride), S2222 (N-Benzoyl-L-isoleucyl-L-glutamyl-glycyl-Larginine-p-nitroaniline hydrochloride), and S2366 (L-Pyroglutamyl-L-prolyl-L-argininep-Nitroaniline hydrochloride) were obtained from Chromogenix. Thrombin generation assays were performed using the calibrated automated thrombography (CAT) method using a Fluoroscan Ascent fluorometer (Thermo Scientific) equipped with a dispenser, (Thrombinoscope BV). H-Gly-Pro-Arg-Pro-OH acetate salt (fibrinolysis inhibiting factor) was from Bachem. Goat polyclonal IgGs against FXII were obtained from Nordic MUbio (cat. number GAHu/FXII). Commercial aPTT agents Pathromtin SL (Siemens), SynthASil, APTT-SP or SynthAFax (HemosIL) or PTT-A, CK PREST, or CEPHASCREEN (Stago) were used. Unless stated otherwise, the buffer used for functional assays contained 20 mM Tris and 50 µM $ZnCl_2$, at pH 7.4.

**Cloning of full-length FXII and FXII deletion mutants**. FXII deletion mutants were engineered by insertion of restriction sites, digestion, and relegation of the F12 cDNA (UniProtKB - P00748 [FA12_HUMAN] and Online Mendelian Inheritance in Man (OMIM) ID: 610619) in pcDNA3 vector. The FXII domain deletion mutants were cloned by PCR-based mutagenesis using the QuikChange Site-directed mutagenesis kit (Stratagene) with pcDNA3-FXII as template. To generate the different mutants, combinations of the primers, as shown in Supplementary Table 1 were used. Plasmid pcDNA3-FXII was digested with HindIII and NotI restriction endonucleases, cDNA fragment was isolated by electrophoresis and was subjected to PCR with the aforementioned primers to insert the EcoRI restriction site. Isolated PCR products were digested by EcoRI, ligated using T4 ligase (New England Biolabs), purified on agarose gel and re-inserted in pcDNA3 vector. DNA sequencing confirmed all plasmid sequences.

**Expression of FXII recombinant mutants**. EcoRI-restriction sites were inserted by site-directed mutagenesis 5′- and 3′-terminal of each domain and the assigned sequence was excised. Transient transfection of vectors coding for FXII_fl and FXII mutants into HEK293 (ATCC: CRL-3216), HepG2 (ATCC: BT-8065), or CHO-K1 (ATCC: CCL-61) cells was done using Lipofectamine 2000 (Thermo Scientific), according to the manufacturer's instructions. 48 h after transfection, cell supernatants were collected and concentrated (Amicon Ultra centrifugal filter, 30 K, Millipore). Transfected cells were lysed with Laemmli buffer. Supernatant and cell lysate were analysed for FXII expression by western blotting with goat polyclonal anti-FXII antibody (1:1000) and horseradish peroxidase (HRP)-coupled anti-goat antibody (1:5000, Chemicon, AP106P).

**Cloning, expression, and purification of MBP-coupled peptides**. EcoRI restriction sites were used to insert the PCR-amplified DNA fragment (obtained with primers described above) encoding for the PR domain N-terminal (PR-I), central (PR-II), and C-terminal (PR-III) domain fragments into pMAL-c5X vectors (New England Biolabs GmbH). Ultracompetent XL10-Gold E. coli were transformed with vectors coding for the maltose-binding protein (MBP)-fused constructs MBP-PR-I, MBP-PR-II, and MBP-PR-III or MBP only. Recombinant protein expression was induced by 0.5 mM isopropylthio-β-D-galactoside (Sigma-Aldrich) at 37 °C for 4 h[61]. Bacteria were harvested by centrifugation, resuspended in column buffer (20 mM Tris HCl, 200 mM NaCl, 1 mM EDTA, 1 mM DTT, pH 7.4) supplemented with 0.5 mg/mL lysozyme and protease inhibitor cocktail (Abcam) and lysed by sonication. Cell lysates were centrifuged (20,000 × g for 20 min at 4 °C) and supernatants were loaded on 5 mL amylose resin column (New England Biolabs). Following washing, bound proteins were eluted with column buffer supplemented with 10 mM maltose. Fractions containing mutants were combined and protein concentrations were determined by the Bradford method.

Coomassie brilliant blue staining assessed protein purity. Western blotting was performed using rabbit polyclonal anti-MBP antibody [1:1000,[62]] and HRP-coupled donkey anti-rabbit antibody (1:5000, Jackson Immunoresearch, AB_2340585).

**Binding of PR mutants to microplate-immobilized polyP**. Long-chain polyP was $Ca^{2+}$-preadsorbed and immobilized onto high-binding polystyrene 96-well plates (Immune 2 HB, Thermo Scientific) using EDAC carbodiimide-mediated covalent coupling[28]. In brief, wells were incubated with polyethylenimine (400 ng/mL) in 0.1 M sodium carbonate–bicarbonate buffer, pH 9.2, overnight at 37 °C. Thereafter, wells were incubated with 200 µL of polyP solution (25 µg/mL polyP in 50 mM EDAC, 77 mM 2-[N-morpholino] ethanesulfonic acid hydrate and 1 mM $CaCl_2$, pH 6) for 4 h. Unbound polyP was removed by 2x washing with 2 M LiCl followed by 2x washing with water. Plates were blocked for 2 h with 5% gelatin in PBS, pH 7.4. MBP-PR-I, MBP-PR-II or MBP-PR-III (200 µL, 50 µg/mL each) were incubated for 1 h at 37 °C in PBS supplemented with 0.6% gelatin and 0.05% Tween. Bound proteins were detected using an anti-MBP antibody (1:1,000), HRP-coupled detection antibodies (1:5,000), and substrate reaction (3,3′,5,5′-tetra-methylbenzidine; Sigma-Aldrich) at an absorbance wavelength of 650 nm.

**Pulldown experiments**. FXII_fl or FXII_ΔPR-III (10 µg each) was incubated with kaolin (500 µg/mL) for 10 min. Suspension was centrifuged (1000 g, 1 min at 4 °C) and the pellet was washed three times with cold PBS. Input and pellet were analysed under reducing conditions by western blotting using polyclonal anti-FXII antibodies.

**Generation of FXII_fl or FXII_ΔPR-III transgenic $F12^{-/-}$ mice**. $F12^{-/-}$ mice[63] were backcrossed for more than 10 generations into a C57BL/6 background. For hydrodynamic tail vein injection 50 µg pLIVE plasmids coding for FXII_fl or FXII_ΔPR-III, were diluted in 0.9% saline equivalent to 10% of the body weight of each mouse. The $F12^{-/-}$ mice were anesthetized with isoflurane and the total volume was injected intravenously within 4-7 sec[64,65]. Generally, mice used were 7–13-week-old and were injected 1 week before the collagen/epinephrine challenge. Blood was obtained following day 7 post-injection and pooled plasma was used for clotting analyses.

**$FeCl_3$-induced arterial thrombosis model**. Male mice of 6-12 weeks of age were subjected to a model for arterial thrombosis[30] approved by the "Regierung von Unterfranken" (Germany). Mice were kept with a 12 h light/12 h dark cycle and researchers and technicians did not enter the mouse room during the dark cycle. Temperatures were constantly ~19-21 °C with 40-60% humidity. In brief, WT and $F12^{-/-}$ mice, some of them reconstituted with FXII_fl or FXII_ΔPR-III [8 µg/g body weight, i.v. injection[66]], were anesthetized and the mesentery was externalized through a midline abdominal incision to expose mesenteric arterioles. Animals received fluorescently labelled platelets (1 ×108) from WT donor mice. After topical application of a filter paper (2 ×1 mm) saturated with 20% $FeCl_3$ for 1 min, thrombus formation was analysed by in vivo fluorescence microscopy of mesenteric arterioles over 40 min or until complete vessel occlusion (cessation of blood flow for >10 min) occurred[17,28,29]. A thrombus was defined as a platelet aggregate >20 µm in diameter.

**Pulmonary thrombosis model**. Mice were anesthetized by intraperitoneal injection of ketamine (120 mg/kg body weight) and xylazine (16 mg/kg body weight) in saline (10 mL/kg body weight). Horm collagen (200 µg/kg body weight, Takeda) was mixed with epinephrine (60 µg kg/kg body weight) and slowly injected into the inferior vena cava[28]. Alternatively, mice reconstituted with mutant FXII proteins, were challenged by i.v. administered kaolin (250 µg/g body weight) or buffer. Animals surviving the challenge for >30 min were considered survivors. After the onset of respiratory arrest and while the heart was still beating or after 30 min for those animals that survived, Evans blue dye (1% in 0.9% saline) was retro-orbitally injected to assess lung perfusion[24]. Lungs were excised and photographed. All mice were treated according to national guidelines for animal care at the animal facilities of University Medical Center Hamburg-Eppendorf and approved by local authorities (TVA #76/16). For animal experiments, 8- to 14-week-old mice of either sex (1:1 ratio) were utilized. All procedures were conducted in accordance with 3Rs regulations.

**Tail bleeding assays**. For bleeding times in WT and $F12^{-/-}$ mice, in the absence or presence of i.v. MBP or MBP-PR-III (8 µg/g body weight each) mice were anesthetized and the mouse tail was transected 3 mm from the tip with a razor blade[30]. The bleeding tail was immersed in a 15 mL test tube containing 12 mL pre-warmed PBS. Bleeding time was recorded as the time to cessation of bleeding for 10 s[67]. Blood loss was quantified by measuring the haemoglobin content of blood collected into PBS. Following centrifugation, the pellet was lysed with lysis buffer (8.3 g/L $NH_4Cl$, 1.0 g/L $KHCO_3$ and 0.037 g/L EDTA) and absorbance of the sample was measured at 575 nm.

**Blood collection.** Human plasma was obtained from healthy volunteers with informed consent. Sampling at the Karolinska University Hospital and the University Medical Center Hamburg-Eppendorf was approved by the Stockholm ethics committee (Regionala Etikprövningnämden) and the Ärztekammer Hamburg (#2322), respectively. All protocols followed are compliant with the World Medical Association (WMA) of Helsinki declaration (https://www.wma.net/policies-post/wma-declaration-of-helsinki-ethical-principles-for-medical-research-involving-human-subjects/). Peripheral venous blood was collected into 3.2% trisodium citrate (9:1 blood-to-citrate ratio). The first 10 mL of sample was discarded. Normal PPP was prepared by two consecutive centrifugation steps, each at 3,000 x g for 10 min each.

**pAb-αGAL14 and pAB-αSPC18 generation and purification.** Anti-PR-III and anti-EGF-I antibodies were raised in rabbits against the synthetic peptides corresponding to human FXII Gly320-Thr333 ([320]GALPAKREQPPSLT[333], Biogenes) and Ser83-Cys102 ([83]SPCQKGGTCVNMPSGPHC[102]) sequences, respectively. Peptides used for immunizations were selected using "The Immune Epitope Database" (http://www.iedb.org). Polyclonal antibodies were immunoselected using a protein G sepharose column followed by a column with immobilized antigens.

**Monoclonal antibodies generation and purification.** Generation of hybridoma cells producing monoclonal antibodies was performed by Biogenes, Berlin (https://www.biogenes.de/custom-antibodies). Briefly, the murine FXII null genotype (C57Bl/6 J background) was crossed into the BALB/c background for 7 generations. $F12^{-/-}$ BALB/c mice were intraperitoneally injected with 25 μg human FXII in Freund complete adjuvant on day 0 and Freund incomplete adjuvant on day 28. A 25 μg booster dose in saline was given at day 70. On day 73, spleens were removed, and lymphocytes were fused with P3X63Ag8.653 myeloma cells using a standard polyethylene glycol-based protocol. Hybridomas were cultured at the University Medical Center Hamburg and the supernatants were tested for their capacity to recognize peptide [320]GALPAKREQPPSLT[333], assessed by an ELISA assay and by immunoblotting of plasma FXII. Hybridomas expressing antibody clones mAb-αPR-III_9, mAb-αPR-III_27, mAb-αPR-III_37, and mAb-αPR-III_43, were subcloned and expanded, before being purified using affinity chromatography with a protein G column (Thermo Fisher Scientific). Epitopes of monoclonal anti-FXII antibodies including mAb-αPR-III_37 mapped to peptide [320]GALPAKREQPPSLT[333] in the human FXII (UniProtKB - P00748 [FA12_HUMAN]) sequence.

**mAb-αPR-III_37 ELISA binding assay.** In total 100 μL of 0.1 μM FXII_pd, dissolved in 0.2 M NaHCO₃ (pH 9.4), was incubated in flat bottom polystyrene 96-well plate overnight at 4 °C. Upon antigen blocking, mAb-αPR-III_37 (100 μL of 0.3 μM) were added. FXII-bound antibodies were visualized using HRP-coupled rat anti-mouse antibody (1:2000, Thermo Fisher Scientific, MA1-34732) and substrate reaction.

**Clotting assays.** Activated partial thromboplastin time (aPTT) was measured using a Kugelkoagulometer (ABW Medizin und Technik GmbH)[7]. Briefly, 50 μL of FXII-deficient human or murine plasma, reconstituted to physiologic FXII levels (375 nM) with FXII_pd or recombinant FXII proteins or buffer control (20 μL), was added to 50 μL Pathromtin SL reagent (Siemens HealthCare Diagnostics). After incubation for 120 s at 37 °C, 30 μL of CaCl₂ (62.5 mM) was added and clotting times were recorded. For the clotting assay with pAb-αGAL14, normal PPP was pre-incubated with buffer, insoluble polyP (10 μg/mL), pAb-αGAL14 (end concentration of 2 μM) or an antibody against VASP (pAb-αControl)[68] before recalcification. For the clotting assay with mAb-αPR-III_37, 50 μL normal PPP was spiked with buffer or mAb-αPR-III_37, in the presence of various concentrations of long-chain polyP followed by recalcification. The Kugelkoagulometer aPTT assays have limited sensitivity at and above a system-specific threshold of 180 s.

**Real-time thrombin generation.** FXII-deficient plasma samples (60 μL) were reconstituted to physiologic FXII concentrations with recombinant FXII mutants or plasma-derived FXII_pd (20 μL each) to physiologic FXII levels. Thrombin formation was initiated using kaolin (1 μg/mL), polyP (10 μg/mL), ellagic acid (2.5 μg/mL), *E. coli* polyP (10 μg/mL) or ionomycin-activated platelets (5 × 10⁷ platelets/mL) in 120 μL reaction mixtures containing 4 μM phospholipids (Thrombinoscope BV), 16.6 mM Ca²⁺ and 2.5 mM fluorogenic substrate (Z-Gly-Gly-Arg-AMC, Thrombinoscope BV)[28]. To study PR-III for its interference with surface triggered coagulation, polyP (5 μg/mL) was pre-incubated with MBP, MBP-PR-I, MBP-PR-II or MBP-PR-III (600 nM) for 30 min at 37 °C prior to measuring thrombin formation in normal PPP. For some experiments, TF (1 pM), FXa (0.4 U/mL) or dabigatran (700 ng/mL) were added. pAb-αGAL14 and mAb-αPR-III_37 activated thrombin formation was analysed in 40 μL normal PPP pretreated with 40 μL of antibodies, kaolin (10 μg/mL), short-chain soluble polyP (10 μg/mL) and/or buffer. Addition of 4 μM phospholipids, 16.6 mM Ca²⁺ and 2.5 mM fluorogenic substrate resulted in a total reaction volume of 120 μL when thrombin generation was started. The control antibody used for experiments with pAb-αGAL14, was against double-stranded RNA. All real time thrombin formation experiments were run in triplicate in normal or factor-deficient platelet–poor human plasma supplemented with 4 μM phospholipids. Thrombin generation was quantified using the Thrombinoscope software package (Version 3.0.0.29) that reported means ± SD.

**FXII cleavage and amidolytic activity assays.** Cleavage of FXII_fl or FXII_ΔPR-III (200 nM) was analysed using concentrated supernatant of HEK293 cells transiently transfected with pcDNA3 vectors coding for the respective proteins. Recombinant proteins were incubated with kaolin (100 μg/mL), soluble short-chain polyP (500 μg/mL), DXS (1 μg/mL) or PKa (30 nM) in buffer (20 mM HEPES, pH 7.4, 100 mM NaCl, 10 μM ZnCl₂) supplemented with 4 μM phospholipids at 37 °C, for up to 4 h. Reducing Laemmli sample buffer was added to stop the reaction and the samples were boiled for 5 min and loaded on 10% SDS-PAGE gels. Western blotting was performed using primary goat polyclonal anti-FXII antibody (1:1000) and HRP-coupled anti-goat antibody (1:5000)[51]. 20 μL of the incubated material was diluted in buffer (1:6) and analysed using the chromogenic substrate S2302 (1 mM) at an absorbance wavelength of 405 nm in a Bio-Kinetics Reader (SpectraMax Plus, Molecular Devices) at 37 °C[69]. Samples activated with PKa were pre-incubated with aprotinin (100 KIU/mL). FXII_fl or FXII_ΔPR-III (200 nM each) was incubated with plasmin (8 U/L) or thrombin (4 U/L) and S2302 substrate conversion was monitored as above.

FXII_pd (200 nM) was incubated with pAb-αGAL14 (1 μM) or pAB-αSPC18 (1 μM) for 3 h at room temperature with buffer or kaolin (10 μg/mL). As control pAb-αControl, a polyclonal antibody against VASP (1 μM) was used. FXII_pd (150 nM), normal PPP or FXII-, PK- or FXI-deficient plasma were incubated with buffer, mAb-αPR-III_27, mAb-αPR-III_37 (750 nM) or mAb-αPR-III_37 Fab fragments for 1 h prior to addition of S2302 or S2366 substrates. Fab fragments from mAb-αPR-III_37, were isolated using the Pierce Fab preparation kit (Thermo Scientific). 20 μL normal PPP was incubated with increasing concentrations of long-chain or short-chain soluble polyP, or DXS, in the absence or presence of mAb-αPR-III_37 (75 nM) for 1 h. pAB-αSPC18 and/or mAb-αPR-III_37 (750 μM each) were incubated in normal PPP for 1 h.

aPTT reagents were diluted 1:200 in buffer before incubation with 40 μL normal PPP. Different concentrations of FVIII, FIX and FXI in plasma were established by mixing factor-deficient plasmas (no detectable levels by immunoprint analyses) with normal PPP. From these mixtures, 10 μL were incubated with mAb-αPR-III_37 (40 nM) or buffer in the presence of CaCl₂ (2.5 mM), phospholipids (0.66 μM) and H-Gly-Pro-Arg-OH acetate salt (3.5 mM) for 30 min, after which S2222 was supplemented, and the measurement was started. 40 μL plasma mixtures (FXI-deficient plasma mixed with normal PPP) were incubated with mAb-αPR-III_37 (150 nM), aPTT reagent (1:100 diluted) or buffer and incubated for 1 h, prior to S2366 supplementation. For the patient study, we excluded one outlier based on the Grubss outlier test (alpha of 10%). Addition of the FXI neutralizing antibody αF2[70] interfered with mAb-αPR-III_37 triggered FXIa signal. For all samples, cleavage of the chromogenic substrate (final concentration of 1 mM) was monitored over time (OD of 405 nm).

**Statistical methods.** Normal distribution was determined by a quantile-quantile plot and data were analysed by unpaired 2-tailed Student's *t* test or, in the case of multiple comparisons, one-way analysis of variance (ANOVA) followed by *post hoc* analysis using Tukey's multiple comparisons test. $R^2$-values were calculated by a non-linear regression curve (agonist *vs.* response, 3 parameters). Prism 6.0 (GraphPad) was used for analysis and values of $P < 0.05$ were considered statistically significant. Data are expressed as mean values ± standard deviation, unless indicated otherwise.

**Reporting summary.** Further information on research design is available in the Nature Research Reporting Summary linked to this article.

## Data availability
The authors declare that the data supporting the findings of this study are available within the article and from the authors upon request. The raw data relative to main and supplementary figures are provided within the source data file. Source data are provided with this paper.

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

## Acknowledgements

Part of the experimental work was performed at the University of Würzburg, Germany. We cordially thank Daniela Urlaub, Würzburg, for excellent technical support. We are grateful to Dr. Matthias Willmanns, Hamburg, for constructive comments. E.X.S. acknowledges funding by the National Institutes of Health (R01 HL137695), the Department of Veterans Affairs (Merit Review Award BX003851), and the Oscar D. Ratnoff Endowed Professorship. T.R. acknowledges the German Research Foundation (grants A11/SFB 877, P6/KFO 306, and B8/SFB 841) for funding.

## Author contributions

M.H., C.N., R.K.M, S.K., K.K. and A.J. performed biochemical studies and in vivo experiments. C.D., M.G., S.R-J. and T.R. designed the study. M.H., C.N., R.K.M., M.F., C.D., G.P., P.K., M.A.F., M.G., J.R.N., L.M.B., S.R.J., O.S., A.S. and E.X.S. provided critical tools and discussed data. R.J.S.P. critically reviewed the manuscript. M.H, C.N., L.M.B., E.X.S. and T.R. wrote the manuscript. All authors reviewed the manuscript and approved

its content. The contents are solely the responsibility of the authors and do not necessarily represent the official views of the NIH, U.S. Department of Veterans Affairs or the United States Government.

## Funding

## Competing interests

Diagnostic use of soluble FXII activating agents that target the PR domain is patent protected by the University of Hamburg and T.R. (EU and U.S. reference numbers EP18736859.2 and US16/622064). All other authors declare no conflict of interest.
