## [Peer Review File · Nature Communications]

Identification of the factor XII contact activation site enables sensitive coagulation diagnosticsREVIEWER COMMENTS

Reviewer #1 (Remarks to the Author):

This is an intriguing study that combines the discovery and identification of the mechanism of activation by which the coagulation factor XII is activated by surfaces, and develops a novel and potentially useful reagent and platform for standardizing FXII-activated clotting of plasma. Only minor comments are appended here that may be useful in helping ensure the rigor and reproducibility of the study.

1) For the mouse studies presented in Figure 5, are plasma samples available to validate that the aPTT was extended for the F12-/- mice and reversed upon infusion of FXI_{fl}, while not reversed for the FXII_{deltaPR-III}?

2) Positive controls should be used where possible; for instance, adding FXa or FXIIa into the experiments presented in Figure 4 to show that these proteases bypass the inhibitory actions of the PR-III sequences; adding hirudin to select experiments as a control; inquiring whether the PR-III sequences prohibit the activation of FXII by FXIa.

3) Mention and reference is made to the use of these reagents for determining and measuring the efficacy of inhibiting FXI activation or activity; thus, can FXIa be doped into select experiments as a positive control and likewise either FXI(a) or FXa inhibitors used as controls to validate the sensitivity and utility of the use of these agents?

4) Was short or long polyP used for Figure 4? Can this be better clarified throughout? Are experiments available using platelet releasate or platelet surface-bound polyP as a stimulator? Or are the experiments available only for synthetic polyP?

5) Claims for reproducibility are made based on Figure 8A, which only contains n=3 data points; this data set should be expanded to demonstrate that the data is normally distributed in order to best showcase the rigor and reproducibility of mAb-alphaPR-III₃₇.

Reviewer #2 (Remarks to the Author):

Comments on Heestemans et al :

In this manuscript the authors focused on the heavy chain of FXII, a factor which has been proposed as an interesting antithrombotic target, and screened a series of mutants in order to identify the domain responsible for FXII interaction with contact negatively charged surfaces. Using clotting times and ETP as readouts they identified a 23 amino acids domain in the C-terminal proline-rich domain, named PR-III, as being required for this interaction. They then developed a polyclonal antibody against a central portion of PRIII and showed its inhibitory activity in the two assay in vitro and an antithrombotic activity in a mouse model. Finally, they used a monoclonal antibody against the same subdomain to develop a novel coagulation assay showing increased sensitivity compared to the available ones. This is a well designed and well performed study which brings new light to an old question on the determinants of the interaction of FXII and which provides convincing data for one essential domain within the proline-rich segment.

Main points :

- Although being a very strong indication that the PR-III domain mediates this interaction, the demonstration relies on indirect assays. A more direct proof that PRIII interacts with relevant contact surfaces would add ground to the demonstration.
- Contact surfaces can be provided by activated cells, such as platelets and endothelial cells. An assay comprising such physiologically relevant surfaces would also benefit the story.
- In view of the different results in the literature testing new antithrombotics when performed in different models it would be safe to add another thrombosis model such as laser induced thrombosis.
- Targeting FXII being proposed as preserving normal hemostasis would ask for testing the compound in bleeding time assays.

Minor points :

- The choice of the immunizing peptide, which is shorter than the PR-III domain, is not explained
- The functional assay has been performed with a polyclonal antibody and not with one of the monoclonal antibodies, could the authors explain why ?

- It is suggested that mAb- α PR-III_37 is a surface mimicking antibody, is there more direct evidence for that, what are the key residues on PR-III for its binding ?
- Knowing that FXII binds via its PR-III domain it would be interesting to evaluate binding of selected FXII deletion mutants to activated platelets

Reviewer #3 (Remarks to the Author):

FXII, the first element of the contact pathway, is involved in both hemostasia and inflammation. Although data from patients and animal models revealed that FXII is not relevant for a physiological hemostasia, there are emerging results supporting a key antithrombotic potential for FXII deficiency. Moreover, variants of FXII have been involved in a type of life-threatening inherited swelling disorder, hereditary angioedema (FXII-HAE). Unfortunately, there is still limited information on FXII, particularly how FXII recognizes surfaces leading to its activation. This study has done an exhaustive dissection of FXII using a recombinant model to identify a small stretch of residues that is essential for FXII activation. The 23-residues of the proline-rich region are crucial for FXII activation by different triggering factors. Moreover, a peptide with this sequence competed with surface-induced FXII activation and coagulation. Finally, an antibody against this region induced FXII activation with the subsequent thrombin generation. These findings in addition to fully characterize a key functional domain of this protease, also allow the development of an antibody-based aPTT, which may allow standardizing this universal test.

Main points

- 1) The recombinant FXII deletion variants have been generated in HEK293 human cells. As the glycan composition is dependent of the cell line used to produce the recombinant variant, and glycosylation plays a crucial role in the interaction of FXII with negatively charged surfaces (variants lacking O-glycosylation at the Pro-rich region, mainly Thr309Lys, are involved in FXII-HAE), I strongly recommend to validate the main results of this study with recombinant proteins produced in other cell line.
- 2) Authors must explain the unexpected sizes for some variants in Figure 1. FXII_ΔFib1 has similar size than FXII_{fl}, despite of lacking 42 residues. The MW of FXII_ΔKR is smaller than the expected, probably because this variant also lacks one N-glycosylation.
- 3) The aPTT and thrombin generation results obtained after reconstituting FXII-deficient plasma with different variants strongly support a key role for the C-term of the Pro-rich region in contact activation of FXII. However, the absence of any other domain seems to play a negligible role not only in contact activation, but also in thrombin generation, which is contradictory with previous studies.
- 4) FXII_ΔPR-III has increased sensibility to PKa-cleavage. Time dependent results are shown, but dose-dependent experiments must be done to show the minor dose of PKa required for activation of FXII with null or defective PR-III.
- 5) If the peptide containing the PR-III (Gln317-Ser339) is able to block thrombin generation from normal plasma when activated by PolyP, without affecting activation by PKa, this peptide might constitute an excellent new antithrombotic drug. Have authors any information concerning this issue in models of thrombosis?
- 6) The effect of the polyclonal antibody against the PR-III region is not fully convincing. The chromogenic and thrombin generation assays rendered nice results. However, its effects on the clotting time are much milder. Authors must show the cleavage of FXII in plasma treated with this antibody. Similar studies should be done for the strongest monoclonal antibody (mAb- α PR-III_37), and the Fab-fragments.
- 7) The mAb precisely and reproducibly detect very low levels of FXI (1%). These studies should be validated using plasma from patients with different congenital FXI deficiencies, particularly to show any evidence of the potential predictive value of the bleeding risk in these patients. Unfortunately, I think that the new assay will not solve this problem.
- 8) The role of thrombin or plasmin to facilitate the activation of FXII_ΔPR-III that is speculated in the discussion should be tested.

Minor points.

Use FXII-HAE instead of HAE type III

Reviewer #4 (Remarks to the Author):

The authors presented a study in which they screened FXII for sequences required for contact activation and found that a continuous stretch of 23 amino acids (Gln317-Ser339, designated PR-III) within the FXII proline-rich domain is essential for surface-induced FXII activation and coagulation. In addition, they constructed a recombinant PR-III peptide and observed that it competed with full-length FXII for surface triggered activation. They also demonstrated that mutants lacking PR-III were defective in sustaining platelet-triggered thrombosis in mice. Additionally, they found that antibodies raised against PR-III recapitulated the effects of a surface and induced FXII activation. Based on these findings, they decided to use the FXII activating antibodies to establish coagulation assays that offered novel diagnostic methods.

This manuscript resumes a lot of work and in the view of this reviewer should be better presented in two distinct papers:

1. On the experiments describing the activation of FXII
2. On the novel coagulation assay based on the use of the antibodies directed against the PR-III region of FXII.

Regarding my comments, please find below some questions or request for further details.

- The study appears to be well designed and the approach is logical. Nevertheless, the authors should provide the exact clotting time of the kaolin aPTT in presence of the mutant lacking the PR region. Namely, they provide the aPTT clotting time for the deficient plasma (i.e. 427 sec) while they only report (>180 sec for the PR mutant. If the difference is not of the same amplitude, would this suggest that other sites of FXII are also sensitive to the negatively charged particles? Their results are particularly strange because depleting PR-II and PR-III or PR did not provide the same prolongation of the aPTT than depleting the PR-III region. Did the authors have an explanation for this observation?
- Did the authors test other activators of the contact pathway like ellagic acid or silica? Knowing that aPTT is also dependent on the phospholipid content, did the authors test whether this has an impact on their experiments? If the experiments cannot be replicated with other activator, this may suggest that kaolin activates PR-III region but that other activator may trigger FXII activation via other regions.
- What was the plasma used for FXII-deficient experiments? Did the authors test different plasma to show the robustness of their findings?
- The use of kaolin to trigger the coagulation is not usual. As this reagent is formed of particles, does this have an impact on thrombin generation?
- The authors mentioned total thrombin (did they want to mention endogenous thrombin potential?)
- Avoid the use of generic terms like "slightly faster" etc. Only provide the data.
- Can we really consider polyP as surfaces? Were these experiments confirmed by studies on medical device material (e.g. catheter?)
- The fact that the authors did not succeed in replicating the experiments of Clark et al. is questionable. Can the authors elaborate on that?
- Why did the authors want to use complex antibody assays for FXI inhibitors monitoring? In addition, thrombin may also activate FXI and this is not considered in the present assay. The use of thrombin generation induced via the intrinsic pathway may appear a more suitable solution.

Overall, all the experiments are of great interest for the community but I would definitely recommend to the authors to separate the assay part which could sound like a commercial approach while the other experiments are really interesting for the understanding of contact activation of factor XII. However, to show the robustness of these results, I would recommend that the authors expand their findings to other activators than kaolin and repeat the experiments with different types of activators (as the reagent they reported in the M&M - but were not discussed in the main text).

RESPONSE TO REVIEWERS

IDENTIFICATION OF THE FACTOR XII CONTACT ACTIVATION SITE ENABLES SENSITIVE COAGULATION DIAGNOSTICS (reference number NCOMMS-20-22387)

Edited text in the manuscript is marked in yellow and the original reviewer comments are designated in gray.

Reviewer #1 (Remarks to the Author):

This is an intriguing study that combines the discovery and identification of the mechanism of activation by which the coagulation factor XII is activated by surfaces, and develops a novel and potentially useful reagent and platform for standardizing FXII-activated clotting of plasma. Only minor comments are appended here that may be useful in helping ensure the rigor and reproducibility of the study.

We would like to thank the reviewer for the enthusiastic assessment of our work.

1) For the mouse studies presented in Figure 5, are plasma samples available to validate that the aPTT was extended for the $F12^{-/-}$ mice and reversed upon infusion of FXII_{fl}, while not reversed for the FXII_{ΔPR-III}?

For the data shown in original Figure 5, mice were infused with recombinant FXII and FXII mutants, anesthetized, subjected to major surgery (open abdomen and excavation of the small intestine), and finally challenged by chemical injury that triggered endothelial damage leading to mesenteric vessel thrombosis. It is established that trauma and anesthesia lead to coagulation activation. Due to these limitations, we originally did not collect blood from challenged mice.

However, in this revised submission we performed additional *in vivo* thrombosis assays at reviewer's # 2 suggestion. Within this setting, $F12^{-/-}$ mice were reconstituted by intravenously infused recombinant or transgenic expressed full length FXII (FXII_{fl}) or FXII_{ΔPR-III} variant prior to a thrombotic challenge (please see reviewer 2, point 3, page 6 below for details on these mouse models). Blood was collected by cardiac puncture at the conclusion of experiments defined as, following anesthesia for 35 min or immediately once cardiac arrest ensued. We successfully obtained >150 μL citrated blood in 12/21 challenged mice. However, 8 samples clotted during plasma preparation prior to analysis. Despite the obvious pre-activation, addition of the FXII activator kaolin, shortened the clotting time in FXII_{fl} reconstituted $F12^{-/-}$ mouse plasma (by 30% compared to non-reconstituted $F12^{-/-}$ plasma in a ball coagulometer (Kugelkoagulometer) assay, but was inactive in shortening clotting times in FXII_{ΔPR-III} reconstituted $F12^{-/-}$ plasma.

Additionally, we confirmed that recombinant FXII_{fl}, but not FXII_{ΔPR-III}, has the capacity to normalize defective contact-initiated clotting in $F12^{-/-}$ mouse plasma. We collected blood from non-challenged $F12^{-/-}$ mice, prepared platelet poor plasma, spiked the samples with FXII_{fl} or FXII_{ΔPR-III}, and measured kaolin-triggered clotting times in reconstituted samples (**Figure R1**). Addition of FXII_{fl} normalized the prolonged clotting time to levels similar to wild type mouse plasma ($F12^{-/-}$ plasma with FXII_{fl}: 32 ± 4 s vs. wild type plasma 29 ± 3 s). In contrast, addition of FXII_{ΔPR-III} did not normalize contact-initiated coagulation ($F12^{-/-}$ plasma with FXII_{ΔPR-III}: 92 ± 18 s vs. $F12^{-/-}$ plasma 96 ± 19 s).

Figure R1: FXII_{fl} but not FXII_{ΔPR-III} restores contact-initiated clotting times in F12^{-/-} mouse plasma. Citrated F12^{-/-} mouse plasma was reconstituted with FXII_{fl} or FXII_{ΔPR-III} (375 nM each) and clotting was activated with kaolin in recalcified samples. Clotting time of kaolin-activated recalcified plasma obtained from buffer-infused wild type and F12^{-/-} mice, was included as control, n=3. ** P<0.01.

These murine data are consistent with data generated using human FXII deficient plasma that is reconstituted with FXII_{ΔPR-III} and FXII_{fl} (**Figures 2A, 2B**). The studies noted above have been added in the manuscript on pages 8/9.

2) Positive controls should be used where possible; for instance, adding FXa or FXIIa into the experiments presented in Figure 4 to show that these proteases bypass the inhibitory actions of the PR-III sequences; adding hirudin to select experiments as a control; inquiring whether the PR-III sequences prohibit the activation of FXII by FXIa. As suggested, we added positive controls in key experiments:

- (i) We now show that PR-III did not affect tissue factor (TF)-triggered thrombin formation (**Supplemental figure S3D**). The data demonstrated that the inhibitory action of PR-III does not involve the (i) the TF-driven extrinsic pathway of coagulation, (ii) potential FXIa-mediated FXII activation (FXIa is generated via the feedback activation loop in TF activated plasma), and (iii) the common pathway of coagulation.
- (ii) We initiated thrombin formation in PPP by the addition of FXa, in the presence of MBP-PR-III or MBP. The experimental design allowed for analyses of potential PR-III effects on FXa-downstream reactions and on FXIa-mediated FXII activation, as FXIa is directly generated by thrombin formed via the feedback-activation-loop. The new data showed that PR-III sequence had no measurable effect on FXa-triggered coagulation, indicating that PR-III inhibits reactions upstream of FXa and does not affect FXIa-mediated FXII activation (**Supplemental figure S3E**).
- (iii) MBP-PR-III had no measurable effect on FXIIa-triggered plasma clotting times (47 ± 6, 49 ± 7 and 45 ± 6 s in MBP-PR-III, MBP (3.5 μM each) and buffer supplemented PPP. Data show that inhibitory activity of PR-III did not affect FXIIa-driven reactions and were added to the text (page 8).
- (iv) Real time thrombin formation assays showed that hirudin and dabigatran interfered with polyP-initiated coagulation, independently of the presence or absence of PR-III, arguing against PR-III activity beyond coagulation. Data for dabigatran are shown in **Supplemental figure S3F**.
- (v) We confirmed that targeting the surface binding site (PR-III) on zymogen FXII with mAb-αPR-III₃₇ leads to FXIIa activity in plasma (**Figure 7**). Enzymatic activity induced by mAb-αPR-III₃₇ was blocked by FXIIa inhibitor corn trypsin inhibitor (CTI) to background levels. These new data confirm the specificity of mAb-αPR-III₃₇ in inducing FXIIa generation (**Figure R2**, revised manuscript page 10).

Figure R2: CTI inhibits FXIIa enzymatic activity induced by mAb-αPR-III_37 in plasma. PPP spiked with or without CTI (2μM) was supplemented with buffer or mAb-αPR-III_37 (150 nM, molar ratio 1:1) and incubated for one hour. FXIIa formation was measured by conversion of the chromogenic substrate S2302 at 180 min. *** $P < 0.001$.

We added the information that PR-III activity is independent on FXIIa-, FXa-, or TF-driven reactions to the text and **Supplemental figure S3, D-F**. Thank you for the suggestions.

3) Mention and reference is made to the use of these reagents for determining and measuring the efficacy of inhibiting FXI activation or activity; thus, can FXIa be doped into select experiments as a positive control and likewise either FXI(a) or FXa inhibitors used as controls to validate the sensitivity and utility of the use of these agents?

We appreciate the reviewer's attention to detail. Indeed, original Figure 8D showing a possible diagnostic application of mAb-αPR-III_37 for measuring FXI levels required revision. We inadvertently omitted to include the value for undetectable plasma FXI levels (<1%, complete FXI deficiency). Data are now added to the revised **Figure 8D**, allowing the assessment of the specificity of the assay. As suggested, we analyzed FXI levels in samples obtained from patients taking FIIa and FXa inhibitors (dagibatran and apixaban, respectively) and found FXI levels to be within normal range. As advised, we also tested FXI inhibitors. Addition of the FXI neutralizing antibody αF2², interfered with mAb-αPR-III_37 triggered FXa signal. Furthermore, we confirmed that mAb-αPR-III_37 specifically triggers FXIIa formation in plasma using the specific FXIIa inhibitor, CTI (**Figure R2**). Together, the new data validate the specificity and sensitivity of the assay.

Finally, to further determine if mAb-αPR-III_37 can add diagnostic value in screening of FXI deficiency, we collaborated with Dr. Ophira Salomon, Tel Aviv, Israel and obtained samples of FXI deficient patients. It is widely accepted that bleeding tendency in FXI individuals does not correlate with FXI plasma levels. Thus, it constitutes a priority to develop a method to help assess constitutive bleeding risk in these patients. In these studies, we found that activation of FXI deficient patient samples with mAb-αPR-III_37, allowed for discrimination of bleeders and non-bleeders. For further data please refer to reviewer 3, question 7 (page 12/13, **Supplemental figure S8**).

4) Was short or long polyP used for Figure 4? Can this be better clarified throughout? Are experiments available using platelet releasate or platelet surface-bound polyP as a stimulator? Or are the experiments available only for synthetic polyP?

We apologize for not being precise. We have divided our response regarding polyP in sections.

Regarding the polyP chain length, we originally used synthetic, long chain insoluble polyP (100- to 400-units chain length) as the FXII contact activator. In the revised manuscript, we specify the polyP used. Mostly long chain insoluble polymer is used. In instances where short chain polyP is used, we ensured that a clear indication is made in the respective figure legend (**Figures 3, C-D**). Of note, both short chain

(Figures 3C, 3D, 7K, Supplemental figure S6D) and long chain polyP (Figures 2C, 2E, 4, 7K and Supplemental figure S6C) initiated FXII activation in a PR-III dependent manner.

Regarding the origin of polyP, in the original submission, we only used synthetic polyP (ICL Pharmaceutical) as FXII contact activators. To test whether natural polyP similarly activated FXII in a PR-III dependent manner, we purified polyP from living *E. coli*. As expected, natural polyP failed to initiate coagulation in FXII deficient plasma. Reconstitution of FXII-deficient plasma with FXII_{fl} allowed bacterial polyP to generate thrombin. In contrast, in FXII deficient samples reconstituted with FXII_{ΔPR-III}, natural polyP-triggered thrombin generation was significantly defective. Together, these new data show that natural polyP requires PR-III for initiation of FXII contact activation, similar to synthetic polymers. We added this information to **Supplemental figures S2A and S2B** in the revised manuscript.

Regarding the origin of platelet surface polyP: Real time microscopy visualized Ca²⁺-rich polyP microparticles that were retained on activated platelets and function as FXII contact activators on the plasma membrane³. Independent data confirmed polyP located on platelet surfaces⁴. We analyzed FXII_{ΔPR-III} or FXII_{fl} for activated platelet triggered plasma coagulation. In brief, activated platelets readily initiated thrombin formation in the presence of FXII_{fl}, but were inactive at triggering coagulation in FXII-deficient plasma substituted with FXII_{ΔPR-III}. The data confirm the essential role of the PR-III sequence for platelet-driven FXII contact activation. We would like to refer this reviewer to our response to reviewer # 2, remark 2 for additional information (page 5 of this revision letter). Activated platelet polyP-driven contact activation was added to **Figure 5D**.

5) Claims for reproducibility are made based on Figure 8A, which only contains n=3 data points; this data set should be expanded to demonstrate that the data is normally distributed in order to best showcase the rigor and reproducibility of mAb-αPR-III₃₇.

We should have clarified in our original Methods section that although the original data in **Figure 8A** showed the mean of n=3 samples, the assay on these samples was performed 8 times. In the revised manuscript, we added this information to the figure legend of figure 8A. We show all individual data points (n=24) triggered by mAb-αPR-III₃₇ in **Figure R3**.

Figure R3. Pooled data of 8 independent experiments where chromogenic FXIIa activity was measured in PPP following stimulation with mAb-αPR-III₃₇. Absorbance was corrected for buffer levels. Red dots indicate the 3 data points shown in the original figure 8A. n=24.

FXII contact activation in normal PPP by mAb-αPR-III₃₇ leads to a robust and reproducible signal. Similarly, all other data shown are representative of a series of independent experiments and were performed multiple times to confirm reproducibility.

We followed the reviewer's advice and added the number of independent experiments performed to the figure legends.

Reviewer #2 (Remarks to the Author):

Comments on Heestemans et al :

In this manuscript the authors focused on the heavy chain of FXII, a factor which has been proposed as an interesting antithrombotic target, and screened a series of mutants in order to identify the domain responsible for FXII interaction with contact negatively charged surfaces. Using clotting times and ETP as readouts they identified a 23 amino acids domain in the C-terminal proline-rich domain, named PR-III, as being required for this interaction. They then developed a polyclonal antibody against a central portion of PRIII and showed its inhibitory activity in the two assay in vitro and an antithrombotic activity in a mouse model. Finally, they used a monoclonal antibody against the same subdomain to develop a novel coagulation assay showing increased sensitivity compared to the available ones. This is a well designed and well performed study which brings new light to an old question on the determinants of the interaction of FXII and which provides convincing data for one essential domain within the proline-rich segment.

We are grateful for the enthusiastic assessment of our work.

Main points:

- Although being a very strong indication that the PR-III domain mediates this interaction, the demonstration relies on indirect assays. A more direct proof that PRIII interacts with relevant contact surfaces would add ground to the demonstration.

We agree with the reviewer that a definite experiment to prove the direct interaction of FXII PR-III with contact-activating surfaces was lacking in the original manuscript. To demonstrate that PR-III mediates FXII surface binding, we performed a pulldown assay using the FXII contact activator kaolin. In brief, FXII_{fl} and FXII_{ΔPR-III} were incubated with kaolin, the suspension was pelleted, washed 3x, and bound protein was probed by western blotting using anti-FXII antibodies. Furthermore, we confirmed direct interaction of PR-III with contact activator polyP, by ELISA. The new data show that PR-III mediates FXII surface binding. In addition, we refer this reviewer to our response to the last remark of the reviewer (page 7/8 of this revision letter). We included these data to **Figures 4E, 4F**, and page 8 of the revised manuscript.

- Contact surfaces can be provided by activated cells, such as platelets and endothelial cells. An assay comprising such physiologically relevant surfaces would also benefit the story.

Activated platelets initiate coagulation in a FXII contact activation dependent manner^{3, 5, 6}. To analyze the role of PR-III in platelet-driven contact activation, we compared thrombin formation triggered by activated platelets in FXII deficient plasma reconstituted with FXII_{fl} or FXII_{ΔPR-III}. Addition of activated platelets (25 μM ionomycin⁷) triggered thrombin formation in the presence of FXII_{fl}, however polyP failed to initiate coagulation in the presence of FXII_{ΔPR-III}. The data are consistent with isolated platelet polyP-induced FXII contact activation⁷ and show that FXII PR-III is required for procoagulant platelet-driven coagulation. Furthermore, we now also show that *E. coli* polyP-initiated contact activation of FXII_{fl}, but not of FXII_{ΔPR-III}, supporting that the PR-III sequence drives bacterial FXII contact activation. In our hands activated endothelial cells (ECV304, EA.hy926, HUVEC) were inactive in triggering FXIIa activity. We added the data on activated platelet-driven PR-III dependent FXII contact activation to the main manuscript (**Figure 5D**). The *E. coli* polyP data are shown in **Supplemental figures S2A and S2B**. Thank you for this excellent suggestion.

- In view of the different results in the literature testing new antithrombotics when performed in different models it would be safe to add another thrombosis model such as laser induced thrombosis.

As suggested, we analyzed the role of PR-III for thrombosis in two models of lethal pulmonary embolism:

1. We generated *F12^{-/-}* mice with transgene expression of either FXII_ΔPR-III or FXII_fl using hydrodynamic tail vein injection⁸. We and others have previously shown that *F12^{-/-}* mice or wild type mice treated with FXII inhibitors are protected from collagen/epinephrine lethal pulmonary embolism (PE)^{7, 9, 10, 11, 12}.

Consistent with published data, *F12^{-/-}* mice were protected in the collagen/epinephrine PE challenge. Reconstitution of *F12^{-/-}* mice with FXII_fl restored susceptibility to thrombosis to levels seen in wild type mice (10.7 ± 2.5 vs. 10.2 ± 2.6 min). In contrast, FXII_ΔPR-III-transgenic *F12^{-/-}* mice were protected from activated platelet-driven lethal pulmonary embolism. Injection of Evans Blue dye confirmed the vascular occlusions in lungs of wild type and FXII_fl transgenic *F12^{-/-}* mice. In contrast, lung perfusion remained intact in FXII_ΔPR-III expressing *F12^{-/-}* animals.

2. We also reconstituted *F12^{-/-}* mice with recombinant FXII_ΔPR-III or FXII_fl prior to challenge. Intravenous injection of the contact activator kaolin (250 μg/g BW) triggered lethal pulmonary embolism in FXII_fl reconstituted *F12^{-/-}* and wild type mice. In contrast, *F12^{-/-}* mice reconstituted with buffer or FXII_ΔPR-II, each were protected from contact activation-driven venous thrombosis.

The two new thrombosis models are shown in **Figure 5C** and **Supplemental figure S4A** of the revised manuscript.

- Targeting FXII being proposed as preserving normal hemostasis would ask for testing the compound in bleeding time assays.

As suggested, we analyzed whether targeting FXII impacts hemostasis. We compared the bleeding times and blood loss in wild type and *F12^{-/-}* mice injected with MBP or MBP-PR-III. Targeting FXII contact activation with MBP-PR-III had no impact on bleeding time or blood loss, both in wild type and *F12^{-/-}* mice. Although the data are negative, we consider them of importance and included them in **Supplemental figures S4B and S4C**.

Minor points:

- The choice of the immunizing peptide, which is shorter than the PR-III domain, is not explained.

We selected the specific peptide sequence for immunization by *in silico* immunogenic analysis (IEDB Analysis resource, <http://www.iedb.org>, **Figure R4**). Other publicly available online algorithms (<https://webs.iitd.edu.in/raghava/igpred/index.html>, <http://www.cbs.dtu.dk/services/BepiPred/>, and <https://bioinf.ru/aappred/predict>) consistently identified the core sequence of PR-III. Similarly, we used all of these algorithms to select the peptide sequence derived from the EGF-I domain for immunization (pAB-αSPC18, **Figure 6A** of the original manuscript). The use of the algorithm from IEDB Analysis resource was specified in the Methods section of the revised manuscript (page 21).

Figure R4: *In silico* analysis of immunogenicity of the FXII PR region. Predicted immunogenicity of the FXII PR sequence based on bepipred linear epitope *in silico* analysis. “Score” indicates the level of immunogenicity. The amino acid sequence of the PR domain and PR-I, PR-II, and PR-III regions are indicated. The peptide Gly320-Thr333 used for generation of pAB- α GAL14 and mAb- α PR-III_37 is shown in red.

- The functional assay has been performed with a polyclonal antibody and not with one of the monoclonal antibodies, could the authors explain why?

Both polyclonal and monoclonal antibodies, pAB- α GAL14 and mAb- α PR-III_37, respectively, induced FXIIa formation in chromogenic assays and initiated FXIIa-mediated thrombin formation in plasma (Figures 6D, 7I). pAB- α GAL14 initiated plasma clotting (Figure 6E). However, the procoagulant activity of mAb- α PR-III_37 appeared lower and clotting times varied. For this reason, we chose not to include plasma clotting data for mAb- α PR-III_37 antibody. However, co-application of long chain polyP and mAb- α PR-III_37 shortened clotting time of normal PPP, exceeding levels of polyP alone (revised manuscript page 11).

The antibody data are consistent with a two-step FXII contact activation mechanism: (i) initially polyP binds to the FXII heavy chain outside the PR-III region and this polyP binding leads to exposure of the contact activation site within the PR-III region (Decryption). (ii) pAB- α GAL14/mAb- α PR-III_37 bind to PR-III and initiates zymogen activation. We added the potential two step mechanism to the discussion (pages 14/15).

- It is suggested that mAb- α PR-III_37 is a surface mimicking antibody, is there more direct evidence for that, what are the key residues on PR-III for its binding?

In the current study, we did not analyze the mAb- α PR-III_37 epitope on the amino acid residue level. As indicated in the acknowledgement, we patent protected use of FXII activating agents, such as antibodies. Together with partners from the diagnostic industry, we are presently analyzing mAb- α PR-III_37 binding and epitope structure and we are producing new antibodies against PR-III. Structural analyses of mAb- α PR-III_37 binding and FXII PR-III interaction with surfaces will be the topic of future studies.

- Knowing that FXII binds via its PR-III domain it would be interesting to evaluate binding of selected FXII deletion mutants to activated platelets

Activated platelets expose polyP on their plasma membrane that initiates FXII contact activation³. Using the polyP specific probe PPX_Δ12, we previously detected surface polyP on platelet plasma membranes¹³. On that background, we analyzed FXII PR domain mutants for their interaction with polyP. We established an ELISA assay utilizing immobilized polyP, which revealed that MBP-PR-III but not MBP-PR-II and MBP-PR-I binds to polyP. The new data show a direct and specific interaction of PR-III with polyP and were added to **Figure 4F**. Furthermore, we compared FXII_{fl} and FXII_ΔPR-III activation by activated platelets (**Figure 5D**). Activated platelets triggered activation of FXII_{fl}, but not FXII_ΔPR-III, indicating that PR-III mediates the FXII interaction with procoagulant platelet surfaces. These data are consistent with defective surface binding of FXII_ΔPR-III (**Figure 4E**).

Furthermore, we show that PR-III is essential for polyP-mediated FXII contact activation by complementary experiments:

1. polyP and many other contact activators activate FXII_{fl}, but not FXII_ΔPR-III (**Figures 2, 3A-3F, 5D, Supplemental figure S2, A-F**)
2. PR-III interferes with polyP-mediated thrombin generation (**Figures 4C, 4D**).

Taken together, the data demonstrate that PR-III binds to polyP and is essential for platelet polyP-driven coagulation, with implications for arterial and venous thrombosis (**Figure 5, A-C**).

Reviewer #3 (Remarks to the Author):

FXII, the first element of the contact pathway, is involved in both hemostasia and inflammation. Although data from patients and animal models revealed that FXII is not relevant for a physiological hemostasia, there are emerging results supporting a key antithrombotic potential for FXII deficiency. Moreover, variants of FXII have been involved in a type of life-threatening inherited swelling disorder, hereditary angioedema (FXII-HAE). Unfortunately, there is still limited information on FXII, particularly how FXII recognizes surfaces leading to its activation. This study has done an exhaustive dissection of FXII using a recombinant model to identify a small stretch of residues that is essential for FXII activation. The 23-residues of the proline-rich region are crucial for FXII activation by different triggering factors. Moreover, a peptide with this sequence competed with surface-induced FXII activation and coagulation. Finally, an antibody against this region induced FXII activation with the subsequent thrombin generation. These findings in addition to fully characterize a key functional domain of this protease, also allow the development of an antibody-based aPTT, which may allow standardizing this universal test.

Main points

1) The recombinant FXII deletion variants have been generated in HEK293 human cells. As the glycan composition is dependent of the cell line used to produce the recombinant variant, and glycosylation plays a crucial role in the interaction of FXII with negatively charged surfaces (variants lacking O-glycosylation at the Pro-rich region, mainly Thr309Lys, are involved in FXII-HAE), I strongly recommend to validate the main results of this study with recombinant proteins produced in other cell line.

As requested by the reviewer, we validated our key findings using recombinant FXII mutants expressed in cell lines other than HEK293. We used two distinct expression systems, Chinese hamster ovary cells (CHOs) and HepG2 hepatocytes. Western blot analysis revealed that both CHO- and HepG2-expressed FXII mutants migrated at similar apparent molecular weight as HEK293-produced proteins. Furthermore, mutants that were defective in secretion from HEK293 cells were not released from CHO and HepG2 cells either (**Supplemental figures S1A, S1B**).

To validate our functional data with HEK293 cell expressed FXII mutants, we analyzed CHO-derived FXII variants for contact activation. Chromogenic FXIIa-assay showed that addition of polyP and kaolin to CHO cell-expressed FXII_{fl} initiated enzymatic activity. In contrast, CHO cell-derived FXII_{ΔPR-III} was defective in polyP/kaolin-stimulated contact activation and S2302 hydrolysis. Moreover, we compared FXII_{fl} and FXII_{ΔPR-III} for their capacity to initiate contact-induced coagulation. FXII deficient plasma was reconstituted with CHO cell-expressed proteins and thrombin formation in response to kaolin or ellagic acid was measured. While FXII_{fl} led to robust thrombin generation, the FXII_{ΔPR-III} protein failed to generate thrombin (**Supplemental figures S1C, S1D**). These cumulative results are now included in results section (pages 5/6) and the Material & Methods section, and **Supplemental figure S1**.

2) Authors must explain the unexpected sizes for some variants in Figure 1. FXII_{ΔFib1} has similar size than FXII_{fl}, despite of lacking 42 residues. The MW of FXII_{ΔKR} is smaller than the expected, probably because this variant also lacks one N-glycosylation.

We appreciate the reviewer's attention to detail. We agree with the reviewer that the slightly smaller apparent molecular weight of FXII_{ΔFib-I} as compared to FXII_{fl} was hardly visible in SDS-PAGE. Please note, the calculated molecular mass of protein FXII_{fl} and FXII_{ΔFib1} differs by 4.8 kDa (approximately 5%) of the total mass only. We confirmed the DNA sequence for all FXII constructs and mutants. Furthermore, we

re-evaluated the migration profiles of FXII_{fl} and FXII_{ΔFib-I} in all experiments. Consistent with the original Figure 1, FXII_{fl} and FXII_{ΔFib-I} migrated with highly similar apparent molecular weights, suggesting that the SDS-PAGE system is not sensitive at capturing small differences in high molecular weight glycoproteins. The minor difference in apparent molecular weight of FXII_{fl} and FXII_{ΔFib-I} is more visible in western blot analyses of HepG2 and CHO expressed mutants that used the identical DNA vector constructs (**Supplemental figures S1A, S1B**).

FXII is a complex glycosylated protein (<https://www.uniprot.org/uniprot/P00748>). In addition to known FXII glycosylation sites such as Thr309 mentioned by the reviewer, additional O- and N-glycosylation sites exist on FXII. Indeed, in a recent study a new FXII glycosylation site in the KR domain has been identified¹⁴. The loss of the Asn230 N-glycosylation site in FXII_{ΔKR} offers a rationale for the reduced apparent molecular mass of FXII_{ΔKR}, as compared to FXII_{fl}, that exceeds the calculated protein-deduced mass difference.

In the revised manuscript, we indicated that our gel system has limited sensitivity in discriminating small differences in FXII_{fl} and FXII_{ΔFib-I} apparent molecular mass (page 15), and included that loss of glycosylation in FXII_{ΔKR} based on the newly published study (reference 51; López-Gálvez R, *et al.*) can account for the reduced MW of this variant.

3) The aPTT and thrombin generation results obtained after reconstituting FXII-deficient plasma with different variants strongly support a key role for the C-term of the Pro-rich region in contact activation of FXII. However, the absence of any other domain seems to play a negligible role not only in contact activation, but also in thrombin generation, which is contradictory with previous studies.

Again, a good point raised by the reviewer. Previous studies have shown that the FXII_{ΔFib-II/EGF-I} deletion mutant showed largely increased FXII susceptibility for contact activation¹⁵. Additionally, the EGF-I domain was shown to contain a surface binding site¹⁵, while others mapped the surface binding site to the Fib-II domain¹⁶. In contrast, a FXII mutant spanning the EGF-II, KR and PR domains and the light chain (rFXII-U-like) was shown to bind to surfaces. rFXII-U-like undergoes contact activation, demonstrating that the N-terminus of FXII is dispensable for zymogen contact activation¹⁷.

Our data show that PR-III is essential for surface-driven FXII zymogen activation and targeting the sequence using antibodies is sufficient for triggering FXII contact activation.

We propose a two-step mechanism for FXII contact activation that integrates all previously published work and our current data:

Initially, the interaction between FXII and a surface commences by binding of FXII EGF-I and/or possibly the Fib-II domain. This interaction induces a conformational shift that “decrypts” and exposes the contact activation site in PR-III. In line with an inhibitory/shielding function of the Fib-II/EGF-I domains, a FXII_{ΔFib-II/EGF-I} deletion mutant almost “spontaneously” activated¹⁵. Indeed, for us it was challenging to produce the FXII_{ΔFib-II/EGF-I} mutant in a purely zymogen form (**Supplemental figure S7**), while others reported that about 20% of the protein was activated in the supernatant of transfected cells¹⁵.

While the Fib-II/EGF-I domains appear to regulate the activation threshold of FXII, it is the PR-III region that is required for activation. PR-III contains another surface binding site (**Figures 4E, 4F**) and binding to PR-III is sufficient for FXII contact activation (**Figures 6, 7**). Consistently, MBP-PR-III competes with surface-driven FXII zymogen activation (**Figures 4C, 4D**). In line with an essential role of PR-III for surface binding

and contact activation, a FXII mutant (rFXII.lpc) lacking the 319 N-terminal amino acids of the heavy chain but containing the PR-III region, bound to kaolin and was activated by a surface¹⁷. In conclusion, the previously published data and our current findings are consistent with a two-step model of FXII contact activation comprised of (i) zymogen priming via binding to the Fib-II and/or EGF-I domains that exposes the PR-III and (ii) facilitates binding to PR-III that ultimately leads to FXII activation.

Both polyclonal and monoclonal antibodies raised against the PR-III core sequence activated pure FXII in a buffer system and endogenous FXII in plasma. In contrast, antibodies raised against the EGF-I domain failed to trigger FXII activation (**Figures 6C, 7J**).

To validate the proposed two-step FXII contact activation mechanism, we performed additional experiments:

Anti-EGF-II antibody alone does not initiate contact activation but it amplifies the FXIIa-forming activity of mAb- α PR-III_37 (**Figures 6C, 7J**, also see Reviewer 4, remark 9, page 16 of this revision letter).

Furthermore, we incubated plasma with tiny amounts of contact activators (insoluble and soluble polyP or DXS) at concentrations that were not sufficient to trigger FXIIa formation. Addition of mAb- α PR-III_37 together with contact activators readily triggered FXIIa formation. In contrast, high concentrations of surface contact activators triggered FXII activation independently of mAb- α PR-III_37.

The newly generated data on the proposed mechanism of FXII contact activation are shown in **Figure 7K** and **Supplemental figures S6, C-E**.

4) FXII Δ PR-III has increased sensibility to PKa-cleavage. Time dependent results are shown, but dose-dependent experiments must be done to show the minor dose of PKa required for activation of FXII with null or defective PR-III.

As suggested by the reviewer, we compared the dose dependency of PKa-mediated proteolysis of FXII Δ PR-III and FXII Δ PR-III. Following a 60 min incubation, 90 nM PKa activated FXII Δ PR-III, as indicated by the appearance of the FXIIa heavy chain (HC). In contrast, PKa did not significantly activate FXII Δ PR-III under these conditions (**Figure R5**). We added the information that small concentrations of PKa were sufficient in activating FXII Δ PR-III, but not FXII Δ PR-III (page 7). Thank you for the suggestion.

Figure R5: FXII Δ PRIII has higher susceptibility for PKa-mediated activation compared to FXII Δ PR-III. FXII Δ PR-III (panel A) and FXII Δ PRIII (panel B) were incubated with 3.3, 10, 30 or 90 nM PKa for 60 min at 37°C. Samples were analyzed for zymogen activation, indicated by the appearance of the FXIIa heavy chain by immunoblotting using polyclonal anti-FXII antibodies. FXII: FXII zymogen, HC: FXIIa heavy chain.

5) If the peptide containing the PR-III (Gln317-Ser339) is able to block thrombin generation from normal plasma when activated by PolyP, without affecting activation by PKa, this peptide might constitute an excellent new antithrombotic drug. Have authors any information concerning this issue in models of thrombosis?

The FXIIa blocking antibody Garadacimab™ (synonyms CSL 312, 3F7) is currently tested in phase 2/3 trials against HAE and COVID-19 associated thrombo-inflammation. The antibody has a long half-life, while plasma short peptides are rapidly cleared from the circulation. On that background, targeting FXIIa seems to be superior compared to interfering with zymogen contact activation using a PR-III peptide. Similarly, the peptide FXIIa inhibitor PCK has a short half-life limiting its application¹⁸. To show that PR-III has the capacity to shield contact activators, we performed an ELISA-assay, which confirmed MBP-PR-III binding to polyP (**Figure 4F**).

We added a line about the potential use of MBP-PR-III for anticoagulation in the discussion section (page 13).

6) The effect of the polyclonal antibody against the PR-III region is not fully convincing. The chromogenic and thrombin generation assays rendered nice results. However, its effects on the clotting time are much milder. Authors must show the cleavage of FXII in plasma treated with this antibody. Similar studies should be done for the strongest monoclonal antibody (mAb- α PR-III_37), and the Fab-fragments.

We agree with the reviewer that mAb- α PR-III_37 and pAB- α GAL14 are relatively weak FXII activators, as compared to e.g., kaolin. The two-step FXII contact activation mechanisms described above (page 10/11 of the revision letter) and in the revised manuscript (page 14/15) provides a rationale for this effect. However, as suggested, we confirmed that mAb- α PR-III_37 incubation indeed initiated cleavage of FXII in plasma dose-dependently (**Figure R6**).

Figure R6: Cleavage of FXII by mAb- α PR-III_37. Plasma was incubated with a serial dilution series of mAb- α PR-III_37 for 3h at 37°C. Antibody activated samples were separated by SDS-PAGE under reducing conditions and analyzed for zymogen FXII activation assessed by appearance of the FXIIa heavy chain (HC).

Much to our regret we have ran out of Fab-fragments and polyclonal antibodies and could not further analyze these agents. We added the information that mAb- α PR-III_37 initiated zymogen cleavage to the revised manuscript (pages 10/11).

7) The mAb precisely and reproducibly detect very low levels of FXI (1%). These studies should be validated using plasma from patients with different congenital FXI deficiencies, particularly to show any evidence of the potential predictive value of the bleeding risk in these patients. Unfortunately, I think that the new assay will not solve this problem.

We thank the reviewer for this highly challenging suggestion. Dr. Ophira Salomon (Tel Aviv, Israel) provided samples from FXI deficient individuals (<8% detectable FXI,

determined by aPTT-based clotting assays). FXI plasma levels measured by conventional coagulation assays did not correlate with incidence of bleeding ¹⁹.

Using our novel antibody-based assay (**Figures 8D, 8E**), we measured FXIa signal in the samples of FXI deficient patients. Remarkably, antibody-activated FXIa signal was significantly higher in the bleeder group, compared to non-bleeders. Therefore, the antibody-driven contact activation assay enabled us to discriminate individuals with a bleeding phenotype compared to non-bleeders. Although the underlying mechanisms remain to be established, given that FXIIa has an established role in fibrinolysis and hyperfibrinolysis has been reported as the key driver of bleeding in FXI deficient bleeders ^{20, 21}, it raises the possibility that the new antibody-activated PTT assay may exhibit superior sensitivity in capturing fibrinolysis, as compared to classical surface-activated PTT assays. Although the data are preliminary, we added them as supportive evidence in **Supplemental figure S8**.

8) The role of thrombin or plasmin to facilitate the activation of FXII_{fl} to FXII_{ΔPR-III} that is speculated in the discussion should be tested.

As advised, we analyzed thrombin or plasmin for their capacity to activate FXII_{fl} and FXII_{ΔPR-III}. Plasmin led to a similar degree of activation of FXII_{fl} and FXII_{ΔPR-III} proteins. In contrast, FXII_{ΔPR-III} was better activated by thrombin compared to FXII_{fl}. These results (shown in **Supplemental figures S2G, S2H**) are consistent with previously published data showing that thrombin is a weak activator of full length FXII compared to PKa and plasmin ²².

Minor points.

Use FXII-HAE instead of HAE type III

Corrected as suggested, we replaced with FXII-HAE.

Reviewer #4 (Remarks to the Author):

The authors presented a study in which they screened FXII for sequences required for contact activation and found that a continuous stretch of 23 amino acids (Gln317-Ser339, designated PR-III) within the FXII proline-rich domain is essential for surface-induced FXII activation and coagulation. In addition, they constructed a recombinant PR-III peptide and observed that it competed with full-length FXII for surface triggered activation. They also demonstrated that mutants lacking PR-III were defective in sustaining platelet-triggered thrombosis in mice. Additionally, they found that antibodies raised against PR-III recapitulated the effects of a surface and induced FXII activation. Based on these findings, they decided to use the FXII activating antibodies to establish coagulation assays that offered novel diagnostic methods.

This manuscript resumes a lot of work and in the view of this reviewer should be better presented in two distinct papers:

1. On the experiments describing the activation of FXII
2. On the novel coagulation assay based on the use of the antibodies directed against the PR-III region of FXII.

We appreciate the positive assessment of our work. We also appreciate the suggestion of the reviewer to divide the manuscript in two individual papers. However, since the Editor did not recommend to do this, we preserved all data in the revised manuscript. We feel that the combined manuscript highlights how basic structural work can have translational impact in diagnostics and therapeutics and hope that the presented data better capture the broad readership of *Nature Communications*.

Regarding my comments, please find below some questions or request for further details.

- The study appears to be well designed and the approach is logical. Nevertheless, the authors should provide the exact clotting time of the kaolin aPTT in presence of the mutant lacking the PR region. Namely, they provide the aPTT clotting time for the deficient plasma (i.e. 427 sec) while they only report (>180 sec for the PR mutant. If the difference is not of the same amplitude, would this suggest that other sites of FXII are also sensitive to the negatively charged particles? Their results are particularly strange because depleting PR-II and PR-III or PR did not provide the same prolongation of the aPTT than depleting the PR-III region. Did the authors have an explanation for this observation?

We appreciate the reviewer's attention to detail. Typically, aPTT based clotting assays have low sensitivity in discriminating clotting times above a certain threshold limit. For instance, the Siemens BCS using Actin FS contact activator measures up to 180 s and the Siemens Atellica Coag360 using the identical activator (Actin FS) has a maximum aPTT of 160 s only. This means that in clinical practice all aPTT clotting times above a threshold are reported as "over the limit". Similarly, the ball coagulometer (Kugelcoagulometer), used in the current study, has limited (if any) sensitivity in discriminating clotting times above 180 sec.

We added the precise clotting times for the FXII mutants (page 5) and indicated that aPTT-based assays have limited sensitivity above system-specific threshold levels (page 22).

- Did the authors test other activators of the contact pathway like ellagic acid or silica? Knowing that aPTT is also dependent on the phospholipid content, did the authors test whether this has an impact on their experiments? If the experiments cannot be replicated with other activator, this may suggest that kaolin activates PR-III region but that other activator may trigger FXII activation via other regions.

In the revised manuscript, we added more contact activators including dextran sulfate, silica and ellagic acid. We used a broad array of contact activators including kaolin (a

specific silicate, used in some commercial aPTT reagents such as the Stago – CK PREST, **Figures 2B, 2D, 3A, 3B, 8A**), soluble short chain and insoluble synthetic long chain polyP (**Figures 2C, 2E, 3C, 3D, 4C, 4D, 7K**), natural polyP (**Supplemental figures S2A, S2B**), dextran sulfate (**Figures 3E, 3F; Supplemental figures S2E, S2F**), silica (**Figure 8A**), ellagic acid (**Supplemental figures S1D, S2C, S2D**), and polyclonal and monoclonal antibodies (**Figures 6 and 7**). All activators were used to determine FXII contact activation by the same PR-III dependent mechanism.

We agree with the reviewer that the aPTT depends on phospholipid content. Consistent with published data, we also found that a specific contact activator, such as polyP, was more procoagulant in the presence of phospholipids and clotting times were shorter. However, while phospholipids *accelerate* clotting, they do not affect the mechanistic basis of FXII contact activation or *initiate* coagulation without the addition of a surface. In other words, FXIIa formation induced by various activators is strictly dependent on the source of each activator and dependent on the presence of the PR-III region, independently of the presence or absence of phospholipids. Real time thrombin formation assays were performed in samples spiked with phospholipids according to the manufacturer's instructions (e.g. **Figure 2B**). In contrast, the plasma clotting experiments, e.g. as shown in **Figure 2A**, were performed without added phospholipids. In both systems, contact-initiated clotting/thrombin formation was defective in the absence of PR-III sequence.

- What was the plasma used for FXII-deficient experiments? Did the authors test different plasma to show the robustness of their findings?

Again, a good point raised by the reviewer. In all our current and previously published studies we have used FXII deficient plasma with no detectable FXII protein, as assessed by western blotting. For most experiments, we used commercial plasma obtained from individuals with congenital FXII deficiency from George King from three different LOTs and from at least two different donors. This information was added to the revised manuscript (page 17). Furthermore, we used plasma from two FXII deficient individuals collected in our hospital. We never used FXII depleted PPP as the affinity-column based depletion reduces FXII, but also other factors, such as HMWK, PK and C1inh^{23, 24}. As part of the revision, we have also used plasma from *F12^{-/-}* mice that have no detectable FXII in plasma (reviewer 1, question 1), and data obtained from murine plasma matched findings generated in human FXII deficient plasma.

- The use of kaolin to trigger the coagulation is not usual. As this reagent is formed of particles, does this have an impact on thrombin generation?

We agree with the reviewer that ellagic acid or silica is more commonly used to trigger diagnostic aPTT assays, compared to kaolin-based reagents. However, some commercial aPTT reagents, such as STA CK PREST (Stago) or Roche PTT reagent, use kaolin as FXII contact activator. Furthermore, kaolin is frequently used as contact activator in experimental studies. To experimentally address the importance of particles we analyzed the contact activator high molecular weight dextran sulfate (DXS) that is soluble and does not form particles²⁵. DXS is even used by the company Merck to generate FXIIa from zymogen FXII (cat. number 233493). Soluble DXS initiated cleavage of zymogen FXII_{fl} and initiated formation of FXIIa. In contrast, FXII_{ΔPR-III} was resistant to DXS induced activation (**Figures 3E, 3F; Supplemental figures S2E, S2F**).

The new data show that both soluble and insoluble/particulate contact activators have the capacity for initiating FXIIa formation in a PR-III dependent manner. They also argue against a decisive role of contact activators being particles.

- The authors mentioned total thrombin (did they want to mention endogenous thrombin potential?)

We apologize for not being precise. In the revised manuscript, we consistently use ETP (endogenous thrombin potential, “area under the curve”) when reporting total thrombin formed in a thrombin generation assay.

- Avoid the use of generic terms like "slightly faster" etc. Only provide the data.

Corrected. We omitted quantitative adjectives such as “slightly faster” from the results section in the revised version (page 7).

- Can we really consider polyP as surfaces? Were these experiments confirmed by studies on medical device material (e.g. catheter?)

Recent elegant work from the Maas laboratory using real time microscopy, has shown that polyP forms calcium-rich microparticles. Electron microscopy visualized these particles that are retained on the surface of procoagulant platelets and initiate coagulation in a FXII-dependent manner, with implications for thrombus formation under flow ^{3, 26}. Consistently, using immunofluorescence microscopy, the Conway laboratory has visualized polyP on activated platelet surfaces ⁴. The key distinction involves the form of polyP (in addition to its length) i.e., although soluble short chain polyP has limited activity as a FXII contact activator ²⁷, its FXIIa promoting activity significantly increases in the presence of Ca²⁺ ions which leads to formation of insoluble polyP particles ²⁸.

Classical studies have identified an array of polymers that are capable at inducing FXII contact activation (reviewed in ²⁹). Elegant studies in large animals have shown the clinical importance of polymer-triggered contact activation and demonstrated that targeting FXII protects from catheter induced thrombosis ³⁰.

We added the information that polyP form Ca²⁺-rich particles that are retained on the plasma membrane to the revised manuscript (page 3) and cited the work of Yau *et al.* (reference 17).

- The fact that the authors did not succeed in replicating the experiments of Clark *et al.* is questionable. Can the authors elaborate on that?

Using N-terminal deletion mutants Clark *et al.* showed that (i) the Fib-II domain shielded the FXII activation loop maintaining zymogen quiescence and that (ii) EGF-I domain contains a surface binding site ¹⁵. These findings are fully consistent with our proposed two-step mechanism of FXII contact activation: (i) Surface binding to EGF-I induces a conformational change in the FXII heavy chain that leads to unshielding of the activation loop and exposes the contact activation site located within the PR-III region. Consistently, the FXII_ΔFib-II/EGF-I mutant that lacks the protective N-terminal heavy chain, is more easily activated ¹⁵ and we confirmed that specifically the FXII_ΔFib-II/EGF-I mutant, was generating significant FXIIa activity during the purification process (**Supplemental Figure S7**); (ii) while surface binding to the EGF-I domain increases FXII susceptibility for contact activation, our study shows that binding to PR-III is a requirement for surface-induced FXII activation (**Figures 2, 3, and 5**). Consistently, a PR-III peptide competed with surface-driven FXII zymogen activation (**Figure 4, A-D**) and polyclonal and monoclonal antibodies raised against the PR-III sequence activated FXII in a buffer system and in plasma (**Figures 6 and 7**).

In conclusion, our data is consistent with the data from the study by Clark *et al.* in providing proof for a two-step mechanism that consists of (i) zymogen FXII priming via binding to the Fib-II and/or EGF-I domains that exposes PR-III and (ii) zymogen activation by binding to the PR-III sequence.

To confirm the proposed two-step FXII contact activation mechanism, we produced two antibodies that bind to EGF-II domain (pAB-αSPC18, “step 1”) and PR-III

sequence (mAb- α PR-III_37, "step 2"). These antibodies (i) unshielded the zymogen by binding to EGF-II and (ii) induced FXIIa formation by binding to PR-III. As shown in **Figure 6C**, addition of pAB- α SPC18 alone did not induce FXII activation. However, pAB- α SPC18 significantly amplified mAb- α PR-III_37 activity for FXII contact activation (**Figure 7J**). Consistently, contact activating potential of mAb- α PR-III_37 was largely increased by minute amounts of polyP that were unable to induce FXII contact activation. Surfaces readily bind to EGF-II¹⁵ leading to unshielding of the contact activation site in PR-III. For further information we would like to refer the reviewer to question 3 of reviewer 3.

We added the proposed two-step FXII contact activation mechanism (page 14/15) and FXII activation triggered by two distinct antibodies, to **Figure 7J**.

- Why did the authors want to use complex antibody assays for FXI inhibitors monitoring? In addition, thrombin may also activate FXI and this is not considered in the present assay. The use of thrombin generation induced via the intrinsic pathway may appear a more suitable solution.

We agree with the reviewer that our original statement on the potential use of antibody-activated aPTT as a tool to monitor emerging FXI inhibitors and their effects on clotting and/or thrombin generation was speculative. We have deleted this statement in the revised manuscript. We do need to note that our new antibody-triggered PTT assay is performed in the absence of exogenous phospholipids or Ca²⁺, arguing against thrombin-mediated FXI feedback-activation in this setting.

Regarding a possible diagnostic application of the antibody-activated PTT assay beyond the field of classical contact activator-driven coagulation assays, we analyzed FXIa levels triggered by the antibody in samples of FXI deficient individuals. Independent studies have shown that FXI levels measured by conventional clotting based assays have limited predictive value for distinguishing bleeders from non-bleeders within FXI deficient individuals (reviewed in¹⁹). In short, while an aPTT based assay has proven insensitive at distinguishing bleeders from non-bleeders in a group of individuals with plasma FXI levels of <8%, mAb- α PR-III_37 triggered FXIa signal was significantly higher in bleeders as compared to non-bleeders (**Supplemental figure S8**). These findings clearly need to be confirmed in larger studies, however we consider them of potential interest. For further details regarding antibody-activated PTT analyses in FXI deficient individuals, we would like to refer this reviewer to our response to reviewer 3, question 7.

Overall, all the experiments are of great interest for the community but I would definitely recommend to the authors to separate the assay part which could sound like a commercial approach while the other experiments are really interesting for the understanding of contact activation of factor XII. However, to show the robustness of these results, I would recommend that the authors expand their findings to other activators than kaolin and repeat the experiments with different types of activators (as the reagent they reported in the M&M - but were not discussed in the main text).

We followed the reviewer's suggestion and expanded our experiments to include additional contact activators, including short and long chain polyP, activated platelets, *E. coli*-derived polyP, dextran sulfate (DXS), and ellagic acid. All activators confirmed the critical role of PR-III region for FXII contact activation. In addition, FXII mutants expressed in HEK293, CHO and HEPG2 cells showed an identical activation mechanism.

REFERENCES

1. Sala-Cunill A, *et al.* Plasma contact system activation drives anaphylaxis in severe mast cell-mediated allergic reactions. *J Allergy Clin Immunol* **135**, 1031-1043 e1036 (2015).
2. Renne T, Gailani D, Meijers JC, Muller-Esterl W. Characterization of the H-kininogen-binding site on factor XI: a comparison of factor XI and plasma prekallikrein. *J Biol Chem* **277**, 4892-4899 (2002).
3. Verhoef JJ, *et al.* Polyphosphate nanoparticles on the platelet surface trigger contact system activation. *Blood* **129**, 1707-1717 (2017).
4. Wijeyewickrema LC, *et al.* Polyphosphate is a novel cofactor for regulation of complement by a serpin, C1 inhibitor. *Blood* **128**, 1766-1776 (2016).
5. Castaldi PA, Larrieu MJ, Caen J. Availability of platelet Factor 3 and activation of factor XII in thrombasthenia. *Nature* **207**, 422-424 (1965).
6. Walsh PN, Griffin JH. Contributions of human platelets to the proteolytic activation of blood coagulation factors XII and XI. *Blood* **57**, 106-118 (1981).
7. Muller F, *et al.* Platelet polyphosphates are proinflammatory and procoagulant mediators in vivo. *Cell* **139**, 1143-1156 (2009).
8. Jimenez-Alcazar M, *et al.* Host DNases prevent vascular occlusion by neutrophil extracellular traps. *Science (New York, NY)* **358**, 1202-1206 (2017).
9. Wilbs J, *et al.* Cyclic peptide FXII inhibitor provides safe anticoagulation in a thrombosis model and in artificial lungs. *Nat Commun* **11**, 3890 (2020).
10. Zilberman-Rudenko J, *et al.* Factor XII Activation Promotes Platelet Consumption in the Presence of Bacterial-Type Long-Chain Polyphosphate In Vitro and In Vivo. *Arterioscl Thromb Vasc Biol* **38**, 1748-1760 (2018).
11. Stavrou EX, *et al.* Reduced thrombosis in *Klkb1*^{-/-} mice is mediated by increased Mas receptor, prostacyclin, Sirt1, and KLF4 and decreased tissue factor. *Blood* **125**, 710-719 (2015).
12. Renne T, *et al.* Defective thrombus formation in mice lacking coagulation factor XII. *J Exp Med* **202**, 271-281 (2005).
13. Labberton L, *et al.* A Flow Cytometry-Based Assay for Procoagulant Platelet Polyphosphate. *Cytometry B Clin Cytom* **94**, 369-373 (2018).
14. López-Gálvez R, *et al.* Factor XII in PMM2-CDG patients: role of N-glycosylation in the secretion and function of the first element of the contact pathway. *Orphanet J Rare Dis* **15**, 280 (2020).
15. Clark CC, Hofman ZLM, Sanrattana W, den Braven L, de Maat S, Maas C. The Fibronectin Type II Domain of Factor XII Ensures Zymogen Quiescence. *Thromb Haemost* **120**, 400-411 (2020).
16. Citarella F, te Velthuis H, Helmer-Citterich M, Hack CE. Identification of a putative binding site for negatively charged surfaces in the fibronectin type II domain of

- human factor XII--an immunochemical and homology modeling approach. *Thromb Haemost* **84**, 1057-1065 (2000).
17. Citarella F, *et al.* Control of human coagulation by recombinant serine proteases. Blood clotting is activated by recombinant factor XII deleted of five regulatory domains. *Eur J Biochem* **208**, 23-30 (1992).
 18. Kleinschnitz C, *et al.* Targeting coagulation factor XII provides protection from pathological thrombosis in cerebral ischemia without interfering with hemostasis. *J Exp Med* **203**, 513-518 (2006).
 19. Wheeler AP, Gailani D. Why factor XI deficiency is a clinical concern. *Expert Rev Hematol* **9**, 629-637 (2016).
 20. Gidley GN, Holle LA, Burthem J, Bolton-Maggs PHB, Lin FC, Wolberg AS. Abnormal plasma clot formation and fibrinolysis reveal bleeding tendency in patients with partial factor XI deficiency. *Blood Adv* **2**, 1076-1088 (2018).
 21. Colucci M, *et al.* Reduced fibrinolytic resistance in patients with factor XI deficiency. Evidence of a thrombin-independent impairment of the thrombin-activatable fibrinolysis inhibitor pathway. *J Thromb Haemost* **14**, 1603-1614 (2016).
 22. Ivanov I, *et al.* A mechanism for hereditary angioedema with normal C1 inhibitor: an inhibitory regulatory role for the factor XII heavy chain. *Blood* **133**, 1152-1163 (2019).
 23. Barrowcliffe TW. Standardization of assays of factor VIII and factor IX. *Ric Clin Lab* **20**, 155-165 (1990).
 24. Verbruggen B. Diagnosis and quantification of factor VIII inhibitors. *Haemophilia* **16**, 20-24 (2010).
 25. Tankersley DL, Alving BM, Finlayson JS. Activation of factor XII by dextran sulfate: the basis for an assay of factor XII. *Blood* **62**, 448-456 (1983).
 26. Weitz JI, Fredenburgh JC. Platelet polyphosphate: the long and the short of it. *Blood* **129**, 1574-1575 (2017).
 27. Smith SA, *et al.* Polyphosphate exerts differential effects on blood clotting, depending on polymer size. *Blood* **116**, 4353-4359 (2010).
 28. Donovan AJ, Kalkowski J, Smith SA, Morrissey JH, Liu Y. Size-controlled synthesis of granular polyphosphate nanoparticles at physiologic salt concentrations for blood clotting. *Biomacromolecules* **15**, 3976-3984 (2014).
 29. Vogler EA, Siedlecki CA. Contact activation of blood-plasma coagulation. *Biomaterials* **30**, 1857-1869 (2009).
 30. Yau JW, *et al.* Selective depletion of factor XI or factor XII with antisense oligonucleotides attenuates catheter thrombosis in rabbits. *Blood* **123**, 2102-2107 (2014).

REVIEWER COMMENTS

Reviewer #1 (Remarks to the Author):

No further questions; thorough manuscript.

Reviewer #2 (Remarks to the Author):

The authors satisfactorily answered all my questions and queries and i have no further comment.

Reviewer #3 (Remarks to the Author):

Heestemans et al have significantly improved an outstanding study. They have answered most of my concerns and have supplied further evidences supporting most of their conclusions.

I only have a major concern.

The results supporting the usefulness for the mAb to identify bleeders and non-bleeders among patients with FXI deficiency are surprising and really promising. However, I think that this conclusion is too preliminary with the current status of the research: only 4 patients in each group, without additional information such as the genetic defect, the type of deficiency, and the levels of FXI determined by immunological methods (which are not related to the bleeding tendency in patients with FXI deficiency). Moreover, additional controls such as patients with heterozygous deficiency of FXI and healthy subjects would be required to validate the usefulness of this method for an accurate determination of FXI levels and its prognostic value. According to the source of the patients included in this study, they are probably Ashkenazy patients sharing the same gene variant. Moreover, it is highly probable that all of them have a CRM- deficiency, without FXI in plasma. Additionally, I do not understand why bleeders have increased FXI signal than non-bleeders. I expected the opposite. FXII levels and the activation of FXII in these patients are also required to fully understand the results and get strong conclusions. So, I strongly recommend deleting this result from the study, only suggesting potential usefulness for the mAb.

Javier Corral

Reviewer #4 (Remarks to the Author):

I thank the authors for their revisions. Nevertheless, there are still some aspects of the paper that need to be more discussed. So please find some minor suggestions.

- In the introduction, the authors directly compare aPTT and PT referring to the standardization of the PT with the INR while it is not the case with the aPTT. They mentioned that the aPTT cannot be standardized due to the different activating properties of the aPTT. Nevertheless, batch to batch variability and inter-reagent variability are also encountered with the PT, e.g. the inter-reagent variability observed with the NOACs. This introduction sounds like if the authors would like to find absolutely limitations to the current aPTT without discussing these limitations in an appropriate context. Thus, the criticism against the current aPTT should be better documented providing, for example, missed attempt for standardization.

- As suggested in my previous review, the coagulation times for the different mutants are really (probably statistically) different. Can the authors explain this difference in light of the conclusion they drew (i.e. that only the PR region is involved). One can see that none of these mutants, once added to a FXII deficient plasma led to coagulation times similar to that of the factor deficient plasma in absence of the mutants.

- Can the authors elaborate more on their results showing that depleting PR-II and PR-III seems to still provide a residual activity while depleting PR-III alone provide coagulation time in the same range as having a FXII depleted plasma? This is strange since, if only PR-III would be involved in that process, then, removing PR-III in addition to PR-II would give results in the same range. Here the reported coagulation times are clearly different (198 ± 24 and 442 ± 79 s for PR-II/PR-III vs

PRIII alone depletion in FXII deficient plasma). Although I can acknowledge some variability at high coagulation times, the CV should be more than 100% at 200 seconds (just a bit more than 3 minutes) to consider these results similar.

- I did not find an explanation from the authors regarding the inconsistency of the results between clot-based and real-time TG triggered by kaolin. Namely, one can see that CT is more prolonged with the PR-III condition than with the PRII-PRIII condition but, on the other hand, the TG results show more important inhibition of the ETP in presence of PR-PR-III depletion than with the PR-III alone depletion. Can the authors further elaborate on that? I find this very interesting since the PR-II/PRII deletion provides results similar to HC or complete PR depletion while only PR-III provides different results. This seems to be only the case for the kaolin condition (TG results with polyP provides similar results between PRII/PRIII and PRIII depletion).

- Interestingly, depleted PR-II provides a different TG with probably lower velocity index and higher peak (in addition to higher ETP) with polyP as an activator. Can the authors provide a possible explanation for that result?

- The authors did not address my comments regarding: "The use of kaolin to trigger the coagulation is not usual. As this reagent is formed of particles, does this have an impact on thrombin generation?" Maybe I did not formulate my question correctly. My question was on the potential interference induced by kaolin in the TG well. I suppose this reagent was not added to the calibrator wells of TG. In fact, this is very interesting to observe the difference in the time course of thrombin generation with kaolin and polyP (longer lag time for kaolin but higher peak, lower lag time with polyP but also lower peak, differential effect towards depletion of the PR-II region). This should be further investigated.

- Did the authors test the mutant PRII-PRIII (and also the other PR(x)-depleted mutants) with the other techniques as reported in supplemental figure S2? In addition, the TG curves reported in supplemental figure S2 (panel A & B) are very different from figure 2. What the different between natural and synthetic polyP only the length of is there other possible explanations?

- The source of PolyP is not very clear (sometimes, the authors reported long chain polyP and in other sentences they reported short chain polyP). This should be harmonized to facilitate reading and understanding.

- Last comment: the manuscript is not very easy to follow even though the results are very exciting and promising. Thus, I would recommend that authors try to improve the flow of the idea in order to facilitate reading.

I would like to sincerely thank the authors for their constructive response to my comments.

RESPONSE TO REVIEWERS

IDENTIFICATION OF THE FACTOR XII CONTACT ACTIVATION SITE ENABLES SENSITIVE COAGULATION DIAGNOSTICS (reference number NCOMMS-20-22387A)

Edited text in the manuscript is marked in yellow, original reviewer comments are designated in gray.

REVIEWER COMMENTS

Reviewer #1 (Remarks to the Author):

No further questions; thorough manuscript.

We are pleased that all questions by Reviewer # 1 were satisfactorily addressed in the latest revision of the manuscript.

Reviewer #2 (Remarks to the Author):

The authors satisfactorily answered all my questions and queries and i have no further comment.

We are grateful for the positive assessment of our work.

Reviewer #3 (Remarks to the Author):

Heestemans et al have significantly improved an outstanding study. They have answered most of my concerns and have supplied further evidences supporting most of their conclusions. I only have a major concern.

The results supporting the usefulness for the mAb to identify bleeders and non-bleeders among patients with FXI deficiency are surprising and really promising. However, I think that this conclusion is too preliminary with the current status of the research: only 4 patients in each group, without additional information such as the genetic defect, the type of deficiency, and the levels of FXI determined by immunological methods (which are not related to the bleeding tendency in patients with FXI deficiency). Moreover, additional controls such as patients with heterozygous deficiency of FXI and healthy subjects would be required to validate the usefulness of this method for an accurate determination of FXI levels and its prognostic value. According to the source of the patients included in this study, they are probably Ashkenazy patients sharing the same gene variant. Moreover, it is highly probable that all of them have a CRM- deficiency, without FXI in plasma. Additionally, I do not understand why bleeders have increased FXI signal than non-bleeders. I expected the opposite. FXII levels and the activation of FXII in these patients are also required to fully understand the results and get strong conclusions. So, I strongly recommend deleting this result from the study, only suggesting potential usefulness for the mAb.

We thank the reviewer for the enthusiastic evaluation of our revised manuscript. We agree that accurate determination of FXI levels in patients and its prognostic value requires a comprehensive clinical study and that is beyond the scope of our current manuscript. As suggested by the reviewer, we have deleted the passage and

corresponding supplemental figure S8 from the manuscript's discussion section and, as advised, merely added a brief comment on potential use of the antibody.

Reviewer #4 (Remarks to the Author):

I thank the authors for their revisions. Nevertheless, there are still some aspects of the paper that need to be more discussed. So please find some minor suggestions.

We thank the reviewer for the positive evaluation and the time dedicated in reviewing our revised manuscript. We have addressed the requested minor revisions.

- In the introduction, the authors directly compare aPTT and PT referring to the standardization of the PT with the INR while it is not the case with the aPTT. They mentioned that the aPTT cannot be standardized due to the different activating properties of the aPTT. Nevertheless, batch to batch variability and inter-reagent variability are also encountered with the PT, e.g. the inter-reagent variability observed with the NOACs. This introduction sounds like if the authors would like to find absolutely limitations to the current aPTT without discussing these limitations in an appropriate context. Thus, the criticism against the current aPTT should be better documented providing, for example, missed attempt for standardization.

We apologize for not being precise. Because of the poorly defined surface nature of various FXII contact activators used in aPTT assays, standardization has been challenging. This wide fluctuation in FXII activating potential, in part explains the need for recalibration procedures to determine the normal range and reagent sensitivity when monitoring UFH therapy (Ref. ^{1, 2} and Supplemental Figure S8). Notably, for the reviewer's personal information, in our own hospital changing aPTT reagents (Actin SL to Pathromtin SL), albeit from the same manufacturer (Siemens; https://cdn0.scrvt.com/39b415fb07de4d9656c7b516d8e2d907/1800000007293658/9605b3308401/Siemens-Healthineers-Reagent-Portfolio-brochure_1800000007293658.pdf), required recalibration and adjustment of reference ranges for UFH therapy. Previous attempts to provide an international standard for aPTT using large amounts of FXII contact activators that were distributed among many laboratories had limited success ³. Therefore, we propose that standardization can be facilitated if an antibody-based FXII activation is adopted, in which antibodies are used that can be produced in unlimited amounts, lyophilized for convenient transport and storage, and are (genetically) well-defined.

In the current revision, we omitted any mention on PT/INR standardization and rephrased the sentence on aPTT standardization. Furthermore, we referenced a report on aPTT standardization ⁴.

- As suggested in my previous review, the coagulation times for the different mutants are really (probably statistically) different. Can the authors explain this difference in light of the conclusion they drew (i.e. that only the PR region is involved). One can see that none of these mutants, once added to a FXII deficient plasma led to coagulation times similar to that of the factor deficient plasma in absence of the mutants.

We thank the reviewer for the attention to detail. Work mostly from the Gailani lab has shown that FXII exerts (limited) enzymatic activity even in its zymogen form, providing a possible explanation that addition of FXII mutants to FXII deficient plasma led to coagulation times shorter than that of buffer-spiked FXII deficient plasma. Zymogen FXII has weak activity but enough to generate PKa, which activates FIX independently of upstream factors ^{5, 6}. We have commented on a possible role of FXII zymogen activity. Furthermore, we cannot exclude that a dedicated interplay of altered FXII sensitivity to activation by other plasma proteases (FXIa, PKa, plasmin or others) and “shielding” of PR-III by heavy chain regions underlies the differences between the observed clotting times of the various contact-activation/PR-III deficient mutants. We indicated that zymogen FXII retains some limited activity, providing a rationale for shortening of clotting times as compared to buffer (page 13, middle section).

- Can the authors elaborate more on their results showing that depleting PR-II and PR-III seems to still provide a residual activity while depleting PR-III alone provides coagulation time in the same range as having a FXII depleted plasma? This is strange since, if only PR-III would be involved in that process, then, removing PR-III in addition to PR-II would give results in the same range. Here the reported coagulation times are clearly different (198 ± 24 and 442 ± 79 s for PR-II/PR-III vs PR-III alone depletion in FXII deficient plasma). Although I can acknowledge some variability at high coagulation times, the CV should be more than 100% at 200 seconds (just a bit more than 3 minutes) to consider these results similar.

There is a lack of reliable structural information pertaining to full length FXII, as X-ray data on human (or mouse) FXII that also reflects complex post-translational glycosylation are lacking. Most likely, the high number of proline residues in the PR region enables flexibility of the protein, further complicating structural analysis. In addition, the PR-III region appears to be unique to FXII and homology screening (e.g. using BLAST [<https://blast.ncbi.nlm.nih.gov/Blast.cgi>]) did not identify homologous domains in other proteins. Together, it is challenging to draw definite conclusions regarding the structure of FXII mutants, including the combined PR-II/PR-III and PR-III deletion variants. Interestingly, it is known that PR regions mediate binding (please see ⁷ for additional information) in a process that is presumably involving the specific polyproline helix. Consistently, PR domains involved in binding operate as sensors for negatively charged surfaces. Furthermore we cannot exclude that combined loss of PR-II/PR-III might force a FXII zymogen conformation with enhanced catalytic activity, which does not occur with individual deletion of PR-II or PR-III.

We have added a reference ⁷ (Ref 56 in the revised manuscript) that reviews the functions of proline rich domains in mediating binding, such as interactions with surfaces, that sheds light on the specific structural features and implications on function of these domains/regions.

- I did not find an explanation from the authors regarding the inconsistency of the results between clot-based and real-time TG triggered by kaolin. Namely, one can see that CT is more prolonged with the PR-III condition than with the PR-II-PR-III condition

but, on the other hand, the TG results show more important inhibition of the ETP in presence of PR-PR-III depletion than with the PR-III alone depletion. Can the authors further elaborate on that? I find this very interesting since the PR-II/PRII deletion provides results similar to HC or complete PR depletion while only PR-III provides different results. This seems to be only the case for the kaolin condition (TG results with polyP provides similar results between PRII/PRIII and PRIII depletion).

Both clotting (Fig. 2A) and real time thrombin generation (Fig. 2B-E) assays consistently showed that PR-III is essential for contact-mediated coagulation and fully support the key conclusion of our study. The reviewer is correct that in Fig. 2A, FXII- Δ PR-III seems to be less “coagulant” than FXII- Δ PR-II/PR-III. In contrast, it appears that in Fig. 2B, FXII- Δ PR-III has slightly increased procoagulant activity as compared to FXII- Δ PR-II/PR-III. Despite this, these minor fluctuations were ultimately statistically non-significant and merely a result of experimental variation and the nature of this complex assay. The minor increase in ETP probably arises from artifactual peaks long after normal thrombin generation peak formation has occurred. In support of this, the illustrative thrombogram of kaolin-triggered thrombin generation in Fig. 2D shows the presence of a small, artifactual increase in thrombin generation in the presence of FXII- Δ PR-III, albeit ~20 minutes after thrombin generation triggered by FXII_{fl} and other FXII- Δ PR variants. Although biologically inconsequential, this small increase would be sufficient to raise the mean ETP value for FXII- Δ PR-III, despite its clear lack of meaningful procoagulant activity in response to kaolin. We apologize for the confusion and have added a comment in the discussion (page 15, bottom).

- Interestingly, depleted PR-II provides a different TG with probably lower velocity index and higher peak (in addition to higher ETP) with polyP as an activator. Can the authors provide a possible explanation for that result?

We are impressed by the reviewer’s attention to detail. Indeed, in the real time thrombin generation curves shown in Fig. 2E, FXII- Δ PR-II has a slightly lower velocity index and minor difference in higher peak thrombin. In contrast, using kaolin as the activator (Fig. 2D), this mutant is indistinguishable from FXII_{fl} and FXII- Δ PR-I. However, the data from Fig. 2E represent a typical experiment and overall did not reach statistical significance. A possible explanation for these minor fluctuations might be based on structural alterations in the proline-rich domain with its typical binding motifs (please see reference ⁷ and comments above). Kaolin at the concentration used for this experiment, was a more potent contact activator than long-chain polyP used in this specific experiment (resulting in an ETP of around 1500 vs. 1000 nM*min). When we used crosslinked particulate polyP (Mandrel salt) for thrombin generation, thrombin formation was similar to that observed in the presence of kaolin (particles) and undistinguishable among FXII_{fl}, FXII- Δ PR-I and FXII- Δ PR-II. Hence, in our hands, procoagulant activity of crosslinked polyP was similar to kaolin and exceeded that of long-chain polyP. Together, these data suggest that deficiency in PR-II either increases surface exposure of PR-III or reduces shielding of PR-III by other FXII heavy chain regions, providing a possible explanation for these results.

Due to size restrictions of the journal, we decided not to add the information to the manuscript file. However, similar to all other valuable information, we have asked Nature Communications to publish the reviewers' comments and our responses on their webpage to provide the information for specialized readers.

- The authors did not address my comments regarding: "The use of kaolin to trigger the coagulation is not usual. As this reagent is formed of particles, does this have an impact on thrombin generation?" Maybe I did not formulate my question correctly. My question was on the potential interference induced by kaolin in the TG well. I suppose this reagent was not added to the calibrator wells of TG. In fact, this is very interesting to observe the difference in the time course of thrombin generation with kaolin and polyP (longer lag time for kaolin but higher peak, lower lag time with polyP but also lower peak, differential effect towards depletion of the PR-II region). This should be further investigated.

As suggested, we tested for a possible interference of kaolin particles with real time thrombin generation (fluorometric CAT assay) in the wells. We tested for dose dependency of thrombin generation stimulated by kaolin and long-chain polyP. For comparison, we additionally included the FXII contact activator ellagic acid (Figure 1).

Figure 1: Kaolin, polyP, and ellagic acid induce thrombin generation in a dose-dependent manner. Real time thrombin generation in human platelet poor plasma (PPP) stimulated with increasing concentrations of **(A)** kaolin (0.01 – 100 µg/ml), **(B)** long-chain polyP (0.005 – 50 µg/ml), and **(C)** ellagic acid (0.001 – 1 µg/ml). Normal PPP supplemented with buffer (Tris 20 mM, pH 7.4) was used as negative control. The experiment was performed in triplicate.

The data argue against a fundamental difference of thrombin generation induced by kaolin, polyP, and ellagic acid. Furthermore, the data show that kaolin readily triggers thrombin production dose-dependently, arguing against an interference of particles with the assay. The data are consistent with our own previously published data on FXIIa-triggered thrombin generation (e.g. Fig. 2, ⁸; Fig. 4, ⁹) and data by other independent groups ^{10, 11, 12, 13}.

- Did the authors test the mutant PRII-PRIII (and also the other PR(x)-depleted mutants) with the other techniques as reported in supplemental figure S2? In addition, the TG curves reported in supplemental figure S2 (panel A & B) are very different from figure 2. What the different between natural and synthetic polyP only the length of is there other possible explanations?

In preliminary experiments using chromogenic S2302-based FXIIa assays, we confirmed that FXII_ΔPR-II-PR-III, FXII_ΔHC and FXII_ΔPR were completely defective in contact activation following stimulation by both DXS and ellagic acid. Results were similar to those obtained with FXII_ΔPR-III (supplemental Figure S2). Together, the data consistently showed that PR-III is required for FXII zymogen contact activation.

Regarding any potential differences among synthetic vs. cell purified (natural) polyP: For more than a decade our laboratory has worked with natural polyP and we were the first to show a role of natural polyP for coagulation *in vivo* and to visualize the polymer on procoagulant platelet surfaces. Clearly, there are significant differences when comparing synthetic and natural polyP, among them:

- (i) Dispersity of the polymer chain length of synthetic polyP is low as compared to natural polyP. Polymer synthesis using condensation of phosphoric acid allows for production of rather defined molecules. In contrast, cells produce both short- and long-chain polyP ¹⁴ that differ in their procoagulant potential when analyzed in solution ¹⁵.
- (ii) We did not determine the exact concentration of purified polyP (similarly, kaolin “concentration” has limited importance since the surface properties of the kaolin particles determines their contact activation capacity). Furthermore, at least in yeast, polymer chain length is known to be dependent on energy status of the cells ¹⁶, suggesting that natural polyP is regulated by environmental factors.
- (iii) It is known that multiple proteins bind polyP ^{17, 18}. Binding of exopolyphosphatases interferes with procoagulant polyP activity ⁹. Furthermore, multiple other cationic proteins and other substances block polyP-driven procoagulant reactions ¹⁹. The *E. coli* derived polyP preparation likely contains polyP-binding proteins that modulate function and activity of the polymer.

Each of these three points offers a rationale for the observed procoagulant activity of natural polyP.

- The source of PolyP is not very clear (sometimes, the authors reported long chain polyP and in other sentences they reported short chain polyP). This should be harmonized to facilitate reading and understanding.

As suggested, we specified whether short- or long-chain polyP were used for all experiments shown.

- Last comment: the manuscript is not very easy to follow even though the results are very exciting and promising. Thus, I would recommend that authors try to improve the flow of the idea in order to facilitate reading.

As recommended, we gave our manuscript to a colleague outside the research group and he made correction according to your suggestions, which improved the flow of the manuscript. Furthermore, we deleted duplicate information. To not interfere with the positive assessments of the other reviewers, we refrained from large structural revisions of the manuscript.

I would like to sincerely thank the authors for their constructive response to my comments.

It was our pleasure, we thank you for your valuable time in improving our study.

References

1. Marlar RA, Clement B, Gausman J. Activated Partial Thromboplastin Time Monitoring of Unfractionated Heparin Therapy: Issues and Recommendations. *Sem Thromb Hemost* **43**, 253-260 (2017).
2. Favaloro EJ, Kershaw G, Mohammed S, Lippi G. How to Optimize Activated Partial Thromboplastin Time (APTT) Testing: Solutions to Establishing and Verifying Normal Reference Intervals and Assessing APTT Reagents for Sensitivity to Heparin, Lupus Anticoagulant, and Clotting Factors. *Sem Thromb Hemost* **45**, 22-35 (2019).
3. Poller L. Standardization of the APTT test. Current status. *Scand J Haematol Suppl* **37**, 49-63 (1980).
4. van den Besselaar AM. Standardization of the activated partial thromboplastin time for monitoring of heparin therapy: where should we go? *Ric Clin Lab* **19**, 371-377 (1989).
5. Kearney KJ, *et al.* Kallikrein directly interacts with and activates Factor IX, resulting in thrombin generation and fibrin formation independent of Factor XI. *Proc Natl Acad Sci U S A* **118**, e2014810118 (2021).
6. Noubouossie DF, *et al.* Red blood cell microvesicles activate the contact system, leading to factor IX activation via 2 independent pathways. *Blood* **135**, 755-765 (2020).

7. Williamson MP. The structure and function of proline-rich regions in proteins. *Biochem J* **297** (Pt 2), 249-260 (1994).
8. Larsson M, *et al.* A factor XIIa inhibitory antibody provides thromboprotection in extracorporeal circulation without increasing bleeding risk. *Sci Transl Med* **6**, 222ra217 (2014).
9. Labberton L, *et al.* Neutralizing blood-borne polyphosphate in vivo provides safe thromboprotection. *Nat Commun* **7**, 12616 (2016).
10. Semeraro F, *et al.* Extracellular histones promote thrombin generation through platelet-dependent mechanisms: involvement of platelet TLR2 and TLR4. *Blood* **118**, 1952-1961 (2011).
11. Schlagenhauf A, *et al.* Polyphosphate in Neonates: Less Shedding from Platelets and Divergent Prothrombotic Capacity Due to Lower TFPI Levels. *Front Physiol* **8**, 586 (2017).
12. Furugohri T, Morishima Y. Paradoxical enhancement of the intrinsic pathway-induced thrombin generation in human plasma by melagatran, a direct thrombin inhibitor, but not edoxaban, a direct factor Xa inhibitor, or heparin. *Thromb Res* **136**, 658-662 (2015).
13. van der Meijden PE, *et al.* Dual role of collagen in factor XII-dependent thrombus formation. *Blood* **114**, 881-890 (2009).
14. Weitz JI, Fredenburgh JC. Platelet polyphosphate: the long and the short of it. *Blood* **129**, 1574-1575 (2017).
15. Smith SA, *et al.* Polyphosphate exerts differential effects on blood clotting, depending on polymer size. *Blood* **116**, 4353-4359 (2010).
16. Vagabov VM, Trilisenko LV, Kulaev IS. Dependence of inorganic polyphosphate chain length on the orthophosphate content in the culture medium of the yeast *Saccharomyces cerevisiae*. *Biochemistry (Mosc)* **65**, 349-354 (2000).
17. Rao NN, Gomez-Garcia MR, Kornberg A. Inorganic polyphosphate: essential for growth and survival. *Annu Rev Biochem* **78**, 605-647 (2009).
18. Kornberg A, Rao NN, Ault-Riche D. Inorganic polyphosphate: a molecule of many functions. *Annu Rev Biochem* **68**, 89-125 (1999).
19. Smith SA, Choi SH, Collins JN, Travers RJ, Cooley BC, Morrissey JH. Inhibition of polyphosphate as a novel strategy for preventing thrombosis and inflammation. *Blood* **120**, 5103-5110 (2012).

REVIEWERS' COMMENTS

Reviewer #3 (Remarks to the Author):

The authors have followed my suggestions and they have even improved their manuscript. No further comments.

Reviewer #4 (Remarks to the Author):

Dear authors,

I would like to sincerely thank you for the detailed responses you provided to my list of question. It was a pleasure to review your manuscript. I have no further concerns. Very nice work.

Best regards,